# SiT: Simulation Transformer for Particle-based Physics Simulation

## Abstract

In this paper, we focus on learning-based methods for simulation. Most existing particle-based simulators adopt graph neural networks (GNNs) to model the underlying physics of particles. However, they force particles to interact with all neighbors without selection, and they fall short in capturing material semantics, such as viscosity or plastic deformations, for different particles, leading to unsatisfactory performance, especially in generalization. This paper proposes Simulation Transformer (SiT) to simulate particle dynamics with more careful modeling of particle states, interactions, and their intrinsic properties. Specifically, besides the particle tokens, which encode the states of particles into latent space, SiT generates interaction tokens to extract high-level representations for interactions and selectively focuses on essential interactions by allowing both tokens to attend to each other. In addition, SiT learns material-aware representations by learnable abstract tokens, which will participate in the attention mechanism and boost the generalization capability further. We evaluate our model on diverse environments, including fluid, rigid, and deformable objects, which cover domains of different complexity and materials. Without bells and whistles, SiT shows strong abilities to simulate particles of different materials and achieves superior performance and generalization across these environments with fewer parameters than existing methods. Codes and models will be released.

## 1 Introduction

Particle-based physics simulation is a classic and important topic in computer science. It not only facilitates the exploration of underlying principles in physics, chemistry and biology, but also enables the creation of vivid visual effects such as explosion and fluid dynamic in films and games. Different from traditional simulators, such as grid-based (Guo et al., 2016) and mesh-based (Bronstein et al., 2017) methods, particle-based simulators view a system, which is an example in one domain, as a composition of particles and imitate system changes over time by predicting the changes of particle-wise states according to current particle states and particle interactions, of which the latter represents the influence of action-reaction forces, such as the collisions. Consequently, they follow the same forward process without separately considering different constraints to simulate different domains with varying materials, requiring no domain-specific physical priors. Moreover, since in particle-based simulators the dynamics of a system is modeled by the states of particles and their interactions, they also have the potential to possess a strong generalization ability, where they can estimate the dynamics of a system with varying number and configuration of particles in a more robust manner. After learning the dynamics of fluid water in a sandbox, the same particle-based simulator can be used to simulate a waterfall and a river.

Recent particle-based simulators (Battaglia et al., 2016; Schenck & Fox, 2018; Mrowca et al., 2018; Li et al., 2019; Sanchez-Gonzalez et al., 2020; Ummenhofer et al., 2020) often view a system as a graph, and adopt graph neural network (GNN) (Kipf & Welling, 2016) as the basic network structure. In these attempts, each particle is treated as a node in the graph, with edges linking it to all its neighboring particles, assuming interactions mainly occur between particles that are close to each other. Subsequently, state updates of particles are achieved by combining node features with the summation of edge features. While such a GNN-based formulation obtains satisfying simulation results, it faces two issues that affect efficiency and generality. First, it forces each particle to interact with all its nearby particles without providing a selective mechanism, which leads to computational

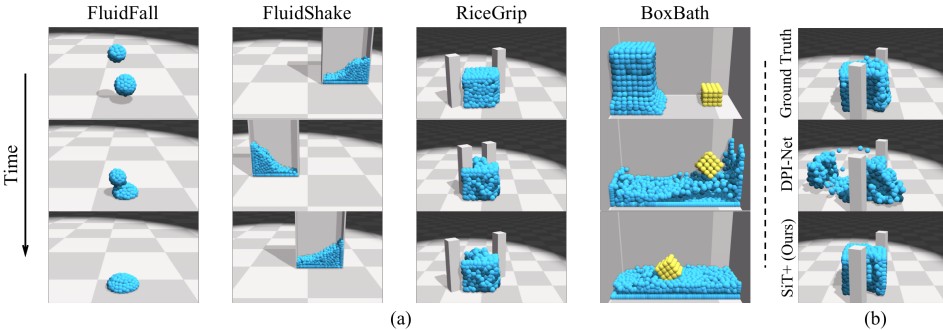

Figure 1: (a). Samples of the datasets. *FluidFall* contains two drops of water. *FluidShake* simulates a block of water in a moving box. *RiceGrip* has a deformable object squeezed by two grippers. *BoxBath* contains a rigid cubic washed by water. (b). Rollouts from generalizations of *RiceGrip*, where we increase the number of particles. SiT with abstract tokens, marked by +, can still maintain the object's shape, while DPI-Net (Li et al., 2019) fail to simulate the deformable object.

redundancy and prevents the discovery of inherent patterns of particle interaction. Second, the GNN-based formulation uses particle-wise attributes to capture both material characteristics, such as viscosity or plastic deformations, and domain-specific semantics, such as the shape of a rigid material. Therefore, it may regard the latter as part of the intrinsic material semantics and fail to generalize to domains with the same materials but different particle amounts and configurations.

In this paper, we propose a novel Transformer-based framework, referred to as Simulation Transformer (SiT), for particle-based physics simulation. The model inherits the powerful multi-head self-attention mechanism in Transformer (Vaswani et al., 2017) to capture particle interactions. To further encourage efficient modeling of complex interactions, instead of treating particle interactions as attention weights obtained by dot-product , we introduce the notion of interaction tokens, which are high-dimensional representations for interactions, to model the rich semantics of particle interactions, such as how the particle is restored after deformations. In addition, to disentangle local material-specific characteristics from global domain-specific semantics, SiT further learns a high-dimensional abstract token for each type of material to capture material semantics, forcing particles of the same material to interact with their corresponding abstract token.

The proposed SiT is more appealing than previous methods in several aspects. First, through capturing particle interactions explicitly with interaction tokens and allowing dynamic inter-token attention, SiT dynamically focuses on essential particle interactions and reduces the computations for redundant and noisy ones. This is crucial for particle-based simulation, especially for domains containing hundreds and thousands of densely placed particles, where the modeling of all particle interactions is redundant, expensive and noisy in practice. Second, thanks to the trainable abstract tokens that disentangle intrinsic material characteristics from domain-specific semantics, we can reuse them to apply SiT to unseen domains of the same materials without retraining. As shown in our experiments, after training on one domain consisting of fluid water and a rigid cubic, SiT still yields fairly faithful simulations when the cubic is replaced with a ball or a bunny comparing with previous work.

To show the effectiveness of SiT, we perform extensive evaluations on four standard environments commonly used in the literature (Li et al., 2019; Sanchez-Gonzalez et al., 2020), covering domains of different complexity and materials. The proposed method achieves superior performance across all these environments with fewer parameters compared to existing methods. We further demonstrate the generalization ability of SiT by adjusting the environments to create new domains and applying SiT to these domains without retraining. In all cases, SiT obtains more realistic simulation results than previous methods, which tend to overfit to the training domains.

## 2  RELATED WORK

**Physics simulation by neural networks**. There are many different kind of representations for physics simulations. Grid-based methods (Lee & You, 2019; Thuerey et al., 2020; Wang et al., 2020) adopt convolutional architectures for learning high-dimensional physical system, while mesh-based simulations (Bronstein et al., 2017; Luo et al., 2020; Hanocka et al., 2019; Nash et al., 2020; Qiao et al., 2020; Weng et al., 2021; Pfaff et al., 2021) typically simulate objects with continuous surfaces, such as clothes, rigid objects, surfaces of water and so on.

Many works (Battaglia et al., 2016; Schenck & Fox, 2018; Mrowca et al., 2018; Li et al., 2019; Sanchez-Gonzalez et al., 2020; Ummenhofer et al., 2020) simulate physics on particle-based systems, where all objects are represented by groups of particles. Specifically, Interaction Network (IN) (Battaglia et al., 2016) simulated interactions in object-level. Smooth Particle Networks (SP-Nets) (Schenck & Fox, 2018) implemented fluid dynamics using position-based fluids (Macklin & Müller, 2013). Hierarchical Relation Network (HRN) (Mrowca et al., 2018) predicted physical dynamics based on hierarchical graph convolution. Dynamic Particle Interaction Networks (DPI-Net) (Li et al., 2019) combined dynamic graphs, multi-step spatial propagation, and hierarchical structure to simulate particles. CConv (Ummenhofer et al., 2020) used spatial convolutions to simulate fluid particles. Graph Network-based Simulators (GNS) (Sanchez-Gonzalez et al., 2020) computed dynamics via learned message-passing. Similar to particle-based systems, COPINGNet (Shao et al., 2021) applies graph networks to simulate rod dynamics, where the discretized rod is the basic unit similar to particle.

Previous work mostly adopted graph networks for simulations and required each particle to interact with all its nearby particles without selective mechanism. In contrast, our SiT employs both particle and interaction tokens and selectively focus on necessary particle interactions through attention mechanism. Experiments show that SiT surpasses existing GNN-based methods and has more robust performances in generalizations.

**Transformer**. Transformer (Vaswani et al., 2017) was designed for machine translation and achieved state-of-the-art performance in many natural langruage processing tasks (Devlin et al., 2019; Radford et al., 2019; Brown et al., 2020). Recently, Transformer starts to show great expandability and applicability in many other fields, such as computer vision (Wang et al., 2018; Carion et al., 2020; Dosovitskiy et al., 2021; Wang et al., 2021; Liu et al., 2021), and graph representations (Zhou et al., 2020; Zhang et al., 2020; Dwivedi & Bresson, 2020). To our knowledge, no attempt has been made to apply Transformer on physics simulation.

Our SiT models the interactions between particles by trainable sub-network given corresponding particle tokens. The same notion of extracting potential semantics between nodes is also applied in Graph Transformer (Dwivedi & Bresson, 2020), which we refer as GraphTrans for short. However, there are differences in our formulations. Specifically, GraphTrans adopts element-wise product between node representations followed by multi-layer perceptron (MLP) to update the interaction embeddings, which store the attention scores in each dimension and are reduced to a scalar for further attention mechanism to update node embeddings. In contrast, our model learns semantic tokens for interactions through sub-network instead of element-wise product. When updating the particle tokens, which are referred as node representations in GraphTrans, both particle and interaction tokens will attend to each other. The interaction tokens are not weighted scores any more. We adopt GraphTrans (Dwivedi & Bresson, 2020) in particle-based simulation and compare it with SiT in experiments. The quantitative results show that SiT achieves better results than GraphTrans.

## 3 METHODOLOGY

### 3.1 PROBLEM FORMULATION

For a particle-based system composed of $N$ particles, we use $\mathcal{X}^t = \{\boldsymbol{x}_i^t\}_{i=1}^N$ to denote the system state at time step $t$, where $\boldsymbol{x}_i^t$ denotes the state of $i$-th particle. Specifically, $\boldsymbol{x}_i^t = [\boldsymbol{p}_i^t, \boldsymbol{q}_i^t, \boldsymbol{a}_i]$, where $\boldsymbol{p}_i^t, \boldsymbol{q}_i^t \in \mathbb{R}^3$ refer to position and velocity, and $\boldsymbol{a}_i \in \mathbb{R}^{d_a}$ represents fixed particle attributes such as its material type. The goal of a simulator is to learn a model $\phi(\cdot)$ from previous rollouts of a system to causally predict a rollout trajectory in a specific time period conditioned on the initial system state $\mathcal{X}^0$. The prediction is run in a recursive manner, where the simulator will predict the state $\hat{\mathcal{X}}^{t+1} = \phi(\mathcal{X}^t)$ at time step $t + 1$ based on the state $\mathcal{X}^t = \{\boldsymbol{x}_i^t\}$ at time step $t$. In practice, we will predict the velocities of particles $\hat{Q}^{t+1} = \{\hat{\boldsymbol{q}}_i^{t+1}\}$, and obtain their positions via $\hat{\boldsymbol{p}}_i^{t+1} = \boldsymbol{p}_i^t + \Delta t \cdot \hat{\boldsymbol{q}}_i^{t+1}$, where $\Delta t$ is a domain-specific constant.

### 3.2 SIMULATION VIA VANILLA TRANSFORMER

To accurately simulate the changes of a system over time, it is crucial to effectively model the interactions among particles, as they indicate the energy transition of a system when constrained by

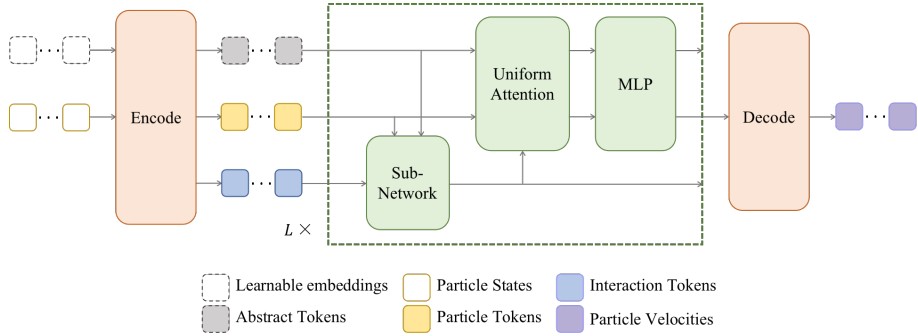

Figure 2: Overview of our transformer-based model with abstract tokens, which are optional. SiT is composed of an encoding layer, stacks of $L$ blocks extended from Transformer encoder blocks, and a decoding layer. Given the particles states at time $t$, the encoding layer will first encode the particles' states and learnable embeddings into particle tokens and abstract tokens, and initialize interaction tokens by particles' states. In each blocks, the sub-network first updates interaction tokens given particle tokens. Then all tokens will dynamically attend to each other via uniform attention and update particle and abstract tokens. An MLP is attached after attention. Finally, the decoding layer will predict the particles' velocities at time $t+1$ given the particle tokens.

material characteristics and physical laws. However, it is infeasible to know a priori how should particles interact with each other. Thus, a selective mechanism is needed to help the simulator focus only on necessary interactions.

Since Transformer (Vaswani et al., 2017) is capable of modeling the dynamical attention scores between tokens via the self-attention module, we can regard the attention weights as the intensity of connectivities and the importance of interactions, where the larger the attention weight the more important the interaction is. Thus, it is naturally a good backbone of building an efficient particle-based simulator. To apply vanilla Transformer in particle-based simulation, we first encode the states of particles into corresponding particle tokens $V = \{\boldsymbol{v}_i^t\}$ by

$$\boldsymbol{v}_i^t = f_V^{\text{enc}}(\boldsymbol{x}_i^t), \tag{1}$$

where $\boldsymbol{v}_i^t \in \mathbb{R}^{d_h}$ is a $d_h$ dimensional vector and $f_V^{\text{enc}}(\cdot)$ is an encoding layer implemented as a MLP. Subsequently, particle interactions are achieved by $L$ blocks of self-attention modules, where in the $l$-th block, particle tokens will attend to each other selectively as:

$$\boldsymbol{v}_i^{l+1,t} = \sum_j \hat{w}_{ij}^v \boldsymbol{v}_j^{l,t}, \tag{2}$$

$$\hat{w}_{ij}^v = \frac{\exp(w_{ij}^v)}{\sqrt{d_h} \cdot \sum_j \exp(w_{ij}^v)}, \tag{3}$$

$$w_{ij}^v = (\boldsymbol{v}_i^{l,t})^\top \boldsymbol{v}_j^{l,t}. \tag{4}$$

Since a system usually contains hundreds of particles and the interactions among particles occur within the neighbors in our settings, considering all possible interactions, of which the number quadratically increases with respect to the number of particles, is computationally redundant and inefficient. Therefore, we follow previous literature Li et al. (2019); Sanchez-Gonzalez et al. (2020) to assume that interactions of distant particles can be omitted, which is realized by a window function:

$$g(\boldsymbol{p}_i^t, \boldsymbol{p}_j^t) = \mathbb{I}\left(||\boldsymbol{p}_i^t - \boldsymbol{p}_j^t||_2 < R\right), \tag{5}$$

where $\mathbb{I}(\text{condition})$ is a indicator function that returns 1 if the condition is satisfied, and $R$ defines the extent of the window. This window function will generate a mask to retain only interactions between neighboring particles as potential candidates in the self-attention modules.

To predict particle states in the next time step, a decoding layer is applied to the updated token of $i$-th particle to obtain its velocity:

$$\hat{\boldsymbol{q}}_i^{t+1} = f_V^{\text{dec}}(\boldsymbol{v}_i^{L,t}), \tag{6}$$

where $f_V^{\text{dec}}(\cdot)$ is implemented by another MLP.

### 3.3 SIMULATION TRANSFORMER (SIT)

Although the vanilla Transformer provides a flexible approach for particle-based simulation, directly applying it leads to inferior simulation results as shown in our experiments. In particular, a vanilla

Transformer uses attention weights, which are scalars obtained via dot-product, to represent particle interactions. A single scalar is insufficient to reflect the rich semantics of particle interactions. For example, in the case of fluid water, when two particles get closer and closer, their interaction will at first act as surface tension to push them towards each other. And when they are sufficiently close, the repulsion between them will dominate the interaction and stop them from further moving towards each other. In addition, since the semantics of one domain is scattered in the tokens of different particles, domain-specific semantics, such as the shape of rigid cubic, are mixed with domain-agnostic semantics, such as the stickiness attribute of materials. In such cases, the former may be misinterpreted as the latter, leading to poor generalization ability. To overcome the limitations of vanilla Transformer, we propose a novel Transformer-based simulator, referred to as Simulation Transformer (SiT), that extends vanilla Transformer with two new types of tokens, namely interactions tokens and abstract tokens. An overview of SiT is included in Figure 2.

**Interaction Tokens.** To capture the rich semantics of particle interactions, instead of a scalar, SiT assigns an interaction token to each potential particle interaction. Specifically, the token for the interaction between particles $\boldsymbol{x}_i^t$ and $\boldsymbol{x}_j^t$ is initialized by

$$\boldsymbol{u}_{ij}^t \quad = \quad f_I^{\text{enc}}(\boldsymbol{x}_i^t, \boldsymbol{x}_j^t), \tag{7}$$

where $\boldsymbol{u}_{ij}^t \in \mathbb{R}^{d_h}$ is a $d_h$ dimensional vector and $f_I^{\text{enc}}(\cdot)$ is the encoding function for interactions.

Subsequently, in each self-attention block SiT will update the interaction tokens as

$$\boldsymbol{u}_{ij}^{l+1,t} \quad = \quad \psi\left(\boldsymbol{v}_i^{l,t}, \boldsymbol{v}_j^{l,t}, \boldsymbol{u}_{ij}^{l,t}, g(\boldsymbol{p}_i^t, \boldsymbol{p}_j^t)\right), \tag{8}$$

where $\psi(\cdot)$ is a learnable sub-network shared for all interaction tokens. Inspired by GNN-based methods (Mrowca et al., 2018; Li et al., 2019; Sanchez-Gonzalez et al., 2020) , SiT adopts the learnable sub-network to enlarge its capacity for handling complex particle interactions. In practice, the sub-network is implemented as

$$\psi\left(\boldsymbol{v}_i^{l,t}, \boldsymbol{v}_j^{l,t}, \boldsymbol{u}_{ij}^{l,t}, g(\boldsymbol{p}_i^t, \boldsymbol{p}_j^t)\right) \quad = \quad \text{MLP}([\boldsymbol{v}_i^{l,t}, \boldsymbol{v}_j^{l,t}, \boldsymbol{u}_{ij}^{l,t}]) \cdot g(\boldsymbol{p}_i^t, \boldsymbol{p}_j^t). \tag{9}$$

As is shown in Figure 6, the update of particle tokens in the self-attention module is thus achieved by

$$\boldsymbol{v}_i^{l+1,t} \quad = \quad \alpha \sum_j \hat{w}_{ij}^v \boldsymbol{v}_j^{l,t} + \beta \sum_k \hat{w}_{ik}^u \boldsymbol{u}_{ik}^{l+1,t}, \tag{10}$$

$$\hat{w}_{ij}^v \quad = \quad \frac{\exp(w_j^v)}{\sqrt{d_h} \cdot \left(\alpha \sum_j \exp(w_j^v) + \beta \sum_k \exp(w_{ik}^u)\right)}, \tag{11}$$

$$\hat{w}_{ik}^u \quad = \quad \frac{\exp(w_{ik}^u)}{\sqrt{d_h} \cdot \left(\alpha \sum_j \exp(w_j^v) + \beta \sum_k \exp(w_{ik}^u)\right)}, \tag{12}$$

$$w_{ij}^v \quad = \quad (\boldsymbol{v}_i^{l,t})^\top \boldsymbol{v}_j^{l,t}, \tag{13}$$

$$w_{ik}^u \quad = \quad (\boldsymbol{v}_i^{l,t})^\top \boldsymbol{u}_{ik}^{l+1,t}, \tag{14}$$

where $\alpha, \beta \in \{0, 1\}$ are the gate controllers, $j$ and $k$ are the indexes of particles within the reception window defined by Equation 5. Equation 13 computes the attention scores between particle tokens, and Equation 14 computes the attention scores between particle and interaction tokens. Subsequently, these attention scores are normalized by Equation 11 and Equation 12. They are then used by Equation 10 to aggregate semantics of particle and interaction tokens, where the gate controllers $\alpha, \beta$ are hyper-parameters that work as balancing coefficients. When $\alpha = 1, \beta = 0$, Equation 10 becomes the original self-attention formula in vanilla Transformer. The multi-head attention version of Equation 10 can be found in Appendix 6.1.2.

It is worth noting that SiT provides a more flexible way for further extensions when compared to vanilla Transformer. Specifically, we can change the value of $\alpha$ and $\beta$ in Equation 10 to adjust the significance of different tokens. And changing the window function in Equation 5 allows SiT to consider different interaction patterns. Finally, by adjusting the learnable sub-network, we can also generalize SiT to consider more complex particle interactions such as interactions beyond a pair of particles and interactions of interactions.

**Abstract Tokens.** To improve the generalization ability of SiT and disentangle domain-specific semantics from its domain-agnostic counterparts, we further apply SiT with material-specific abstract tokens.

For $N_a$ types of materials, SiT adopts $N_a$ abstract tokens $A = \{a_k\}_{k=1}^{N_a}$, each of which is a learnable vector of the same length of particle tokens. Ideally, the abstract token $a_k$ should capture the domain-agnostic semantics of $k$-th material. They act as particle tokens for additional abstract particles, following the same update formula as particle tokens. Therefore with $N_a$ abstract tokens SiT will have $N + N_a$ particle tokens in total: $\{a_1, \cdots, a_{N_a}, x_1^t, \cdots, x_N^t\}$, as is show in Figure 2. Although abstract tokens share the same update formula as particle tokens, unlike particle tokens that decide their potential interactions via the window function in Equation 5, abstract tokens are forced to interact with all particle tokens of its corresponding material, which is achieved by setting the reception field of abstract tokens to the particles of the same material.

Once abstract tokens capture domain-agnostic semantics, they can be reused by SiT when generalizing to domains that have same materials but vary in particle amount and configuration.

### 3.4 TRAINING OBJECTIVE

To train SiT with previous rollouts, the standard mean square error (MSE) loss is applied on the output of our SiT:

$$\text{MSE}(\hat{Q}^{t+1}, Q^{t+1}) \quad = \quad \frac{1}{N} \sum_i ||\hat{q}_i^{t+1} - q_i^{t+1}||_2^2, \tag{15}$$

where $\hat{Q}^{t+1} = \{\hat{q}_i^{t+1}\}_{i=1}^N$ is the estimated velocity , $Q^{t+1} = \{q_i^{t+1}\}_{i=1}^N$ is the ground truth, and $|| \cdot ||_2$ is L2 norm.

While the MSE loss works well for cases with relatively low complexity, in practice some domains often contain multiple types of materials with imbalance numbers of particles. In such cases, the MSE loss will be biased to the material with more particles. Therefore, we further apply a material weighted MSE (WMSE) loss to reduce the effect of imbalance:

$$\text{WMSE}(\hat{Q}^{t+1}, Q^{t+1}) \quad = \quad \frac{1}{K} \sum_k \frac{1}{N_k} \sum_i ||\hat{q}_{i,k}^{t+1} - q_{i,k}^{t+1}||_2^2, \tag{16}$$

where $K$ is the number of material types, $N_k$ is the number of particles belonging to the $k$-th material and $N = \sum_k N_k$, $\hat{Q}_k^{t+1} = \{\hat{q}_{i,k}^{t+1}\}_{i=1}^{N_k}$ and $Q_k^{t+1} = \{q_{i,k}^{t+1}\}_{i=1}^{N_k}$ denote the estimated velocity and ground truth for $k$-th material, and $\hat{Q}^{t+1} = \{\hat{Q}_k^{t+1}\}_{k=1}^K, Q^{t+1} = \{Q_k^{t+1}\}_{k=1}^K$,

## 4 EXPERIMENTS

### 4.1 BASE ENVIRONMENTS

We adopt the environments commonly used in the literature (Li et al., 2019; Sanchez-Gonzalez et al., 2020; Ummenhofer et al., 2020). Samples are displayed in Figure 1. There are four environments in total: *FluidFall* is a basic simulation for two drops of water; *FluidShake* is more complex and simulate the water in a randomly moving box; *BoxBath* simulates the water washing a rigid cubic in fixed box; *RiceGrip* simulates the interactions between deformable rice and two rigid grippers. More details of data generalization can be found in Appendix 6.1.3.

We compare SiT with 4 recent approaches: DPI-Net (Li et al., 2019), CConv (Ummenhofer et al., 2020), GNS (Sanchez-Gonzalez et al., 2020), and GraphTrans (Dwivedi & Bresson, 2020). For fair comparison, we adopt similar training schedules. Quantitative results and model parameters are in Table 5. Qualitative results are in Figure 3. More details can be found in Appendix 6.1.3 and 6.2.2

SiT achieves superior performance on all environments with fewer model parameters. The effectiveness of abstract tokens are more obvious for *RiceGrip* and *BoxBath*, which are more complex. Quantitative results show that SiT can better simulate the dynamics on different environments.

**Comparison with DPI-Net and GNS**. DPI-Net (Li et al., 2019) and GNS (Sanchez-Gonzalez et al., 2020) adopt message-passing graph networks for particle-based simulation. As shown in the results,

Table 1: Quantitative results and model parameters on four environments. SiT achieves superior and reasonable performance with less model parameters on all environments. When adding trainable abstract tokens, which introduce a few more parameters, SiT, marked by +, further improves performance in complex environments. As CConv is designed for fluid dynamics, we report the results on *BoxBath* for reference of the simulation on fluid parts, which is marked by *.

| Methods | FluidFall | | FluidShake | | RiceGrip | | BoxBath | | |
|---|---|---|---|---|---|---|---|---|---|
| | MSE | Params | MSE | Params | MSE | Params | MSE | WMSE | Params |
| DPI-Net | 0.12±0.06 | 0.62M | 1.43±0.52 | 1.02M | 0.11±0.21 | 2.17M | 1.91±0.08 | 1.52±0.33 | 1.98M |
| CConv | 0.08±0.02 | 0.84M | 1.34±0.45 | 0.84M | N/A | N/A | 1.98 ±0.11* | 3.06±0.47* | 0.84M |
| GNS | 0.05±0.02 | 1.59M | 1.45±0.55 | 1.59M | 0.38±0.25 | 1.60M | 1.77±0.87 | 2.62±0.87 | 1.59M |
| GraphTrans | 0.12±0.03 | 0.49M | 8.77±3.97 | 0.96M | 0.36±0.27 | 0.96M | 3.17±0.07 | 2.33±0.47 | 0.97M |
| SiT (Ours) | 0.05±0.02 | 0.40M | **1.08±0.36** | 0.77M | 0.15±0.12 | 0.82M | 1.74±0.08 | 1.42±0.29 | 0.77M |
| SiT+ (Ours) | **0.04±0.01** | **0.40M** | 1.08±0.39 | **0.77M** | **0.07±0.07** | **0.82M** | **1.57±0.06** | **1.39±0.31** | **0.77M** |

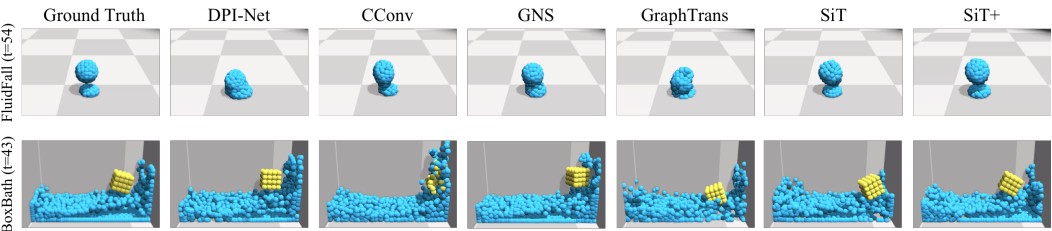

Figure 3: Qualitative results on *FluidFall* and *BoxBath*. When the two drops are getting closer in *FluidFall* but not close enough to merge, previous methods are likely to mix them together due to the newly added neighbors from different drops and incorrect interactions. But SiT can still predict faithful rollouts.

SiT achieves better performance, as the selective mechanism can eliminate the unnecessary interactions, which is further demonstrated by the heat map in Appendix 6.2.1. SiT with abstract tokens further improves the performances especially on *RiceGrip* and *BoxBath*, suggesting the effectiveness of abstract tokens in modeling complex deformations and multi-materials interactions. For *FluidFall*, though the MSEs are close to each other, SiT can better maintain the shape of drops while DPI-Net and GNS fail because they do not have selective mechanism and equally treat the incorrect interactions, which are introduced by the newly added neighbors from different drops when they get closer but not close enough to merge.

**Comparison with CConv**. CConv (Ummenhofer et al., 2020) designs convolutional layers carefully tailored to modeling fluid dynamics, such as an SPH(Monaghan, 1992)-like local kernel, different sub-networks for fluid and boundary particles. Without modeling interactions, CConv propagates information only among particles weighted by distances. On the other hand, SiT assigns high-level tokens for interactions and achieves better results, suggesting the necessity of interaction tokens. Notice that CConv is not suitable for rigid box, we only report result on *BoxBath* for reference of simulation on fluid parts.

**Comparison with GraphTrans** (Dwivedi & Bresson, 2020). The GraphTrans is also a Transformer-based method and updates the interactions using element-wise product from node representations. We represent particles by nodes and adopt layers in GraphTrans as our backbone, which and use the same hidden dimensions and number of blocks. Other settings are completely the same as SiT for better comparison. As is shown in experiments, GraphTrans fails to simulate particles on all cases with much higher errors. This is because our SiT directly models the interactions by interaction tokens generated by sub-network, which is necessary to capture high-level semantics.

## 4.2 GENERALIAZTIONS

To challenge the robustness of our model on more complex settings, we add more particles for *RichGrip* and *FluidShake*, and change the size and shape of rigid object for *BoxBath*. Table 5 shows the upper and lower bound for the number of particles in training set and generalization validation set, as well as the shapes of rigid object.

We mainly compare SiT with DPI-Net, which has fairly good and reasonable performance on all environments. Quantitative results are summarized in Table 5, while qualitative results are depicted in Figure 4 and Figure 1 (b). As is shown, SiT with abstract tokens can better simulate particle dy-

Table 2: MSEs on generalizations. The lists of numbers in *FluidShake* and *RiceGrip* are the range of particles, while the tuples in *BoxBath* denotes number of fluid particles, number of rigid particles, and shape of rigid objects respectively. Training settings are marked by *. We compare SiT with abstract tokens and DPI-Net. SiT+ achieves the best results on all cases.

| Methods | FluidShake [450,627]* | | | RiceGrip [570,980]* | |
|---|---|---|---|---|---|
| | [720,900] | [924,1080] | [1104,1368] | [1060,1345] | [1347,1640] |
| DPI-Net | 2.14±0.36 | 2.78±0.58 | 3.53±0.71 | 2.33±30.64 | 0.87±4.00 |
| SiT+ (Ours) | **1.54±0.37** | **1.90±0.63** | **2.30±0.69** | **0.14±0.10** | **0.17±0.15** |

| Methods | BoxBath (960,64,cubic)* | | | | |
|---|---|---|---|---|---|
| | (1280,64,cubic) | (960,41,bunny) | (960,125,cubic) | (960,136,ball) | (960,120,cuboid) |
| DPI-Net | 4.28±0.11 | 2.49±0.23 | 2.75±0.42 | 2.71±0.35 | 2.97±0.38 |
| SiT+ (Ours) | **2.64±0.11** | **1.72±0.08** | **1.74±0.14** | **1.67±0.13** | **1.92±0.32** |

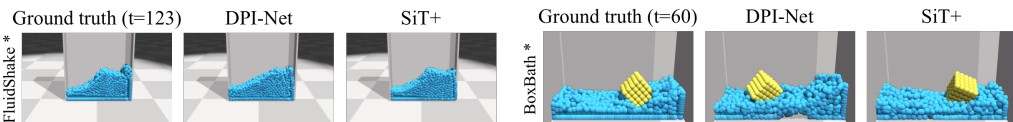

Figure 4: Rendered rollouts on generalized *FluidShake* and *BoxBath*. More particles participate in simulations. SiT with abstract tokens (SiT+) can still faithfully predict the wave of fluid in *FluidShake* and the positions of rigid cubic in *BoxBath*.

namics in generalized environment. suggesting that the abstract tokens and the selective mechanism involving interaction tokens can provide more stable and robust performance. More details about the results and rollouts can be found in Appendix 6.2.3.

## 4.3 ABLATION STUDIES

We comprehensively analyze our SiT and explore the effectiveness of our model in the following ways: (a) the necessity of interaction tokens; (b) the architectures of sub-network, which generates interaction tokens; (c) the values for gate controllers $\alpha, \beta$; (d) the usage of multi-head in uniform attention; (e) the sensitiveness of SiT to radius $R$; (f) the effectiveness of abstract tokens. We conduct our experiments on *FluidShake* for (a) to (e). We verify (f) on *BoxBath* and *RiceGrip*. The quantitative results are in Table 3 and Table 4.

**Necessity of interaction tokens**. We apply vanilla Transformer encoder by configuration A, which uses the same hidden dimension and number of blocks as SiT, and SiT by configuration B, which only changes the output of sub-network to one dimension. As shown in Table 3, both configuration A and B have low performance, suggesting the scalars are insufficient to describe complex interactions.

**Architectures of sub-network**. Configuration C adopts element-wise product between particle tokens, which follows the GraphTrans (Dwivedi & Bresson, 2020); configuration D adopts self-attention blocks, where q is the interaction token from previous layer, k and v are the neighbor particle tokens. As shown in Table 3, configuration C and D have worse performance than SiT, which suggests that the element-wise product is insufficient to model complex interactions, while self-attention block introduces more parameters and complexities, making it harder to train.

**Hyper-paramters for** $\alpha, \beta$. We set $\alpha = 1, \beta = 0$ in configuration E, $\alpha = 1, \beta = 1$ in configuration F, and $\alpha = 0, \beta = 1$ in configuration H, Consequently, both configuration E and H achieve good results, suggesting that the participants of interaction tokens are the key to boost the performance. In practice, we use configuration H for SiT which shows the best results.

**Multi-head uniform attention**. We apply multi-head mechanism in configuration G and set head number as 8. The results show that multi-head mechanism can bring us some improvement. But it also introduces more parameters and consumes lots of memory when computing.

**Sensitiveness to** $R$. Quantitative results are reported on *FluidShake*. As shown in Table 4, SiT is more robust when varying the radius $R$, suggesting the effectiveness of selective mechanism and robustness of SiT.

Table 3: Ablation studies. We comprehensively explore the effectiveness of SiT, including the necessity of interaction tokens, the structure of sub-network, the value of $\alpha, \beta$, and the usage of multi-head in uniform attention. We report MSEs(1e-2) on *FluidShake*, which is a complex environments involving outer forces.

| Configurations | Interaction modeling | $\alpha$ | $\beta$ | Multi-head | FluidShake |
|---|---|---|---|---|---|
| A(Transformer) | Scalar by dot product | 1 | 0 | ✓ | 21.03±10.54 |
| B | Scalar by MLP | 0 | 1 | | 300.43±70.16 |
| C | Tokens by element-wise product | 0 | 1 | | 17.70±6.99 |
| D | Tokens by self attention | 0 | 1 | | 3.17±0.81 |
| E | Tokens by MLP | 1 | 0 | | 14.27±42.08 |
| F | Tokens by MLP | 1 | 1 | | 1.19±0.44 |
| G | Tokens by MLP | 0 | 1 | ✓ | 1.04±0.36 |
| H(SiT) | Tokens by MLP | 0 | 1 | | 1.08±0.36 |

Table 4: Ablation studies on sensitiveness to radius $R$ and abstract tokens. The left parts are MSEs(1e-2) on *FluidShake*, which suggest that SiT is more robust within small range of $R$. Our default setting on all domains is marked by $*$. The right parts are MSEs(1e-2) on *RiceGrip* and *BoxBath*. We replace abstract tokens with dummy tokens, which are fixed value vectors but have same connectivities as abstract tokens.

| Methods | $R = 0.07$ | $R^* = 0.08$ | $R = 0.09$ | Methods | RiceGrip | BoxBath |
|---|---|---|---|---|---|---|
| DPI-Net | 2.60±0.56 | 1.43±0.52 | 1.66±0.48 | SiT w dummy tokens | 2.12±0.46 | 3.98±0.09 |
| SiT | 1.38±0.36 | 1.08±0.36 | 1.37±0.35 | SiT + | 0.07±0.07 | 1.57±0.06 |

**Effectiveness of abstract tokens**. We replace the abstract tokens with dummy tokens, which are randomly initialized vectors with fixed values but have same connectivities as abstract tokens. As shown in 4, SiT with dummy tokens fails on both domains, suggesting the abstract tokens are able to learn the materials semantics and boost SiT's performances.

## 4.4 Few Shots Learning

As shown in Figure 5, we conduct few shots learning experiments on *FluidShake* using SiT without abstract tokens. Even when trained on 60% of data, SiT achieves similar MSEs comparing with DPI-Net trained on 100% of data. When trained with further less data, SiT learns not well enough. As the number of particles differs in each rollout, SiT does not have enough training on all possible cases. On the other hand, the weights are not shared in SiT and would not learn enough on less data comparing with the shared-weight blocks in DPI-Net. SiT is more robust with no less than 60% of the whole training data, but is sensitive to extremely less data.

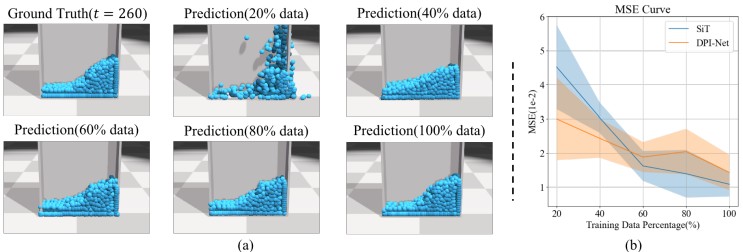

(a)                (b)

Figure 5: Few shots learning on *FluidShake*. Figure (a) shows the rollouts. Figure (b) draws the MSE curves as the number of training examples increases. Our model are robust even with only 60% of training examples.

## 5 Conclusion

In this paper, we propose Simulation Transformer (SiT) and introduce interaction tokens and abstract tokens to simulate domains of different complexity and materials, including hundreds and thousands of particles. Our experimental results show the necessity and effectiveness of interaction and abstract tokens as well as selective mechanism in attention for particle-based simulation. Generalization results further suggest the robustness of SiT. SiT is also flexible for extensions and applications, such as more complex and efficient architectures for sub-network, changing the particle dynamics by attending to alternative abstract tokens with different materials semantics. Finally, SiT makes a successful attempt to apply Transformer into physics simulation and achieve superior performances over existing methods.

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

## 6    APPENDIX

### 6.1    MODEL DETAILS

#### 6.1.1    UNIFORM ATTENTION

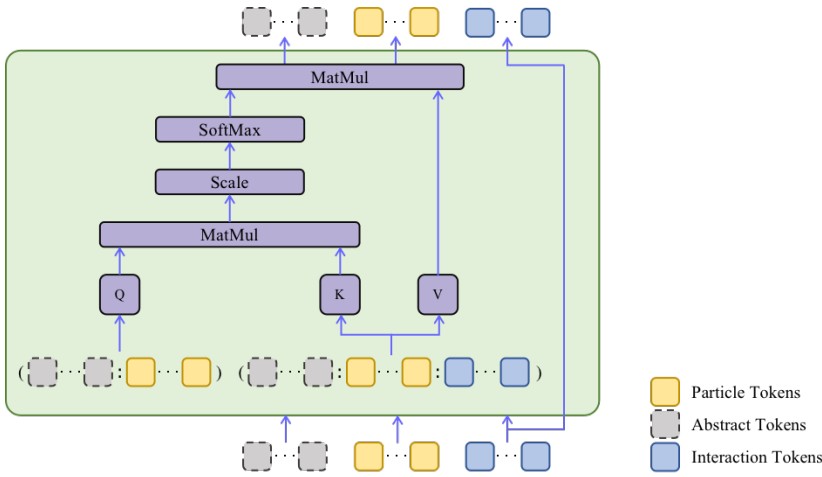

Figure 6: Overview of our uniform attention. The abstract tokens are optional. The tokens will attend to each other and update accordingly.

Figure 6 illustrates how our uniform attention works. Our uniform attention extend self-attention (Vaswani et al., 2017) by additional tokens. The queries are the abstract tokens, which are optional, and particle tokens, while the keys and values are all three types of tokens. Each token will attend to tokens within its reception fields defined by the window function 5. For the abstract tokens, they are forced to attend only to the particles belonging to the same material. The output of uniform attention is the updated tokens except interaction tokens.

#### 6.1.2    MULTI-HEAD UNIFORM ATTENTION

The multi-head version of uniform attention is formulated by

$$
\boldsymbol{v}_i^{l,t} = \boldsymbol{W}^l \left[\mathrm{SA}_1, \mathrm{SA}_2, \cdots, \mathrm{SA}_M\right]^\top, \tag{17}
$$

$$
\mathrm{SA}_m = \alpha \sum_j \hat{w}_{ij}^{m,v}(\boldsymbol{V}^{m,l}\boldsymbol{v}_j^{l-1,t}) + \beta \sum_k \hat{w}_{ik}^{m,u}(\boldsymbol{V}^{m,l}\boldsymbol{u}_{ik}^{l,t}), \tag{18}
$$

$$
\hat{w}_{ij}^{m,v} = \frac{\exp(w_j^{m,v})}{\sqrt{d_h} \cdot \left(\alpha \sum_j \exp(w_j^{m,v}) + \beta \sum_k \exp(w_{ik}^{m,u})\right)}, \tag{19}
$$

$$
\hat{w}_{ik}^{m,u} = \frac{\exp(w_{ik}^{m,u})}{\sqrt{d_h} \cdot \left(\alpha \sum_j \exp(w_j^{m,v}) + \beta \sum_k \exp(w_{ik}^{m,u})\right)}, \tag{20}
$$

$$
w_{ij}^{m,v} = (\boldsymbol{Q}^{m,l}\boldsymbol{v}_i^{l-1,t})^\top(\boldsymbol{K}^{m,l}\boldsymbol{v}_j^{l-1,t}), \tag{21}
$$

$$
w_{ik}^{m,u} = (\boldsymbol{Q}^{m,l}\boldsymbol{v}_i^{l-1,t})^\top(\boldsymbol{K}^{m,l}\boldsymbol{u}_{ik}^{l,t}), \tag{22}
$$

where $m \in 1, 2, \cdots, M$ is the index of head, $W^l \in \mathbb{R}^{d_h \times d_h}$ is the output weight for the concatenation of $M$ heads; $Q^{m,l}, K^{m,l}, V^{m,;}$ are the projection weights of query, key, value for head $m$ at block $l$. The rest symbols are the same with those in uniform attention.

#### 6.1.3    IMPLEMENTATION DETAILS

**Inputs and outputs details**. For *FluidFall*, *FluidShake*, and *BoxBath*, we only use particles' states at time $t$ as inputs and output the velocities at time $t + 1$. For *RiceGrip*, we concatenate particles states

from $t - 2$ to $t$ as inputs and output 6-dim vector for the velocity of the current observed position and the resting position, which is the same setting as DPI-Net for better comparison. For *BoxBath*, we output 7-dim vectors, where 3 dimensions for the predicted velocities, and 4 dimensions for rotation constrains. The rotation constraints are applied only on rigid particles which predict the rotation velocities. We use particles' states at time $t$ as inputs and output the velocities at time $t + 1$ except *RiceGrip*, where we concatenate particles states from $t - 2$ to $t$ as inputs for better comparison with DPI-Net (Li et al., 2019). All states of particles, such as the positions, velocities, and accelerations, are first normalized by mean and standard deviations calculated on corresponding training set.

**SiT details**. We set $R = 0.08$ across all environments. All hidden dimensions in SiT are 128 for default. The MLP after uniform attention has two layers with dimensions 512 and 128. We use 2 blocks shown in Figure 2 for *FluidFall* and 4 blocks for the other three environments. We set $\alpha = 0, \beta = 1$ in our model for default, which show the best results in ablation studies. The learnable embeddings for abstract tokens have the same dimensions as inputs.

**Training**. We train four models independently on these four environments. For common settings, we use Adam optimizer with initial learning rate 0.0008 and plateau scheduler with patience 3 and decreasing factor 0.8. We choose batch size 16 for all environments and train the models for 5 epochs on *FluidShake*, 5 epochs for *BoxBath*, 13 epochs for *FluidFall*, and 20 epochs for *RiceGrip*. No augmentation is involved, such as adding noises to particles states during training. We adopt WMSE loss on *BoxBath* and MSE loss on the rest environments for training.

**Evaluation**. We evaluate the positions between predictions and ground truths by MSE as is shown in equation 15. For *BoxBath*, we also report the WMSE results by equation 16 to better show the performance of simulating different materials.

**Baseline details**. For fair comparison, the following settings are the same with SiT: inputs for models, number of training epochs on different environments, learning rate schedules, and training loss on velocities. On *BoxBath*, all baselines adopt the rigid constraints except CConv, which is designed specifically for fluid dynamics and we use it to mainly compare the fluid simulations. The model-related hyper-parameters for baselines are the same as the original papers, such as the number of message passing is 10 for GNS. Other hyper-parameters for baselines are first chosen the same as their original papers, and then fine-tuned within a small range of changes. For example, in terms of the initial learning rate, 0.0001 works better on GNS while others adopt 0.0008.

## 6.2 EXPERIMENTS

### 6.2.1 ADDTIONAL ABLATION STUDIES

We also visualize the attention scores in each block for a specific particle on *BoxBath* in Figure 7. The red dot in the center is the given particle. Other dots are the neighors. The red particle will attend selectively to its neighbors rather than treating them equally. Different layers focus on different aspect of semantics. Specifically, the particle will focus mainly on closer neightbors in the first layer as is shown in Figure 7 (b).

### 6.2.2 BASE ENVIRONMENTS

**Dataset details**. We use the same setting for our datasets as mentioned in previous work (Li et al., 2019). *FluidFall* contains two fluid drops with different sizes. The size for each drop are randomly generated with one drop larger than the other. Position and viscosity for drops are randomly initialized. This environment contains 189 particles with 121 frames for each rollout. There are 2700 rollouts in training set and 300 rollouts in validation set. *FluidShake* simulates the water in a moving box. The speed of the box is randomly generated at each timestamp. In addition, the size of the box and the number of particles are various for different rollouts. In basic training and validation sets, the number of particles changes from 450 to 627. This environment has 301 frames for each rollout. There are 1800 rollouts in training set and 200 rollouts in validation set. *RiceGrip* contains two grippers and a sticky rice. The grippers' positions and orientation are randomly initialized. The number of particles for rice changes from 570 to 980 with 41 frames for each rollout in training and validation sets. There are 4500 rollouts in training set and 500 rollouts in validation set. *BoxBath* simulates a rigid cubic washed by water in a fixed container. The initial position of fluid block and rigid cubic are randomly initialized. This environment contains 960 fluid particles and 64 rigid par-

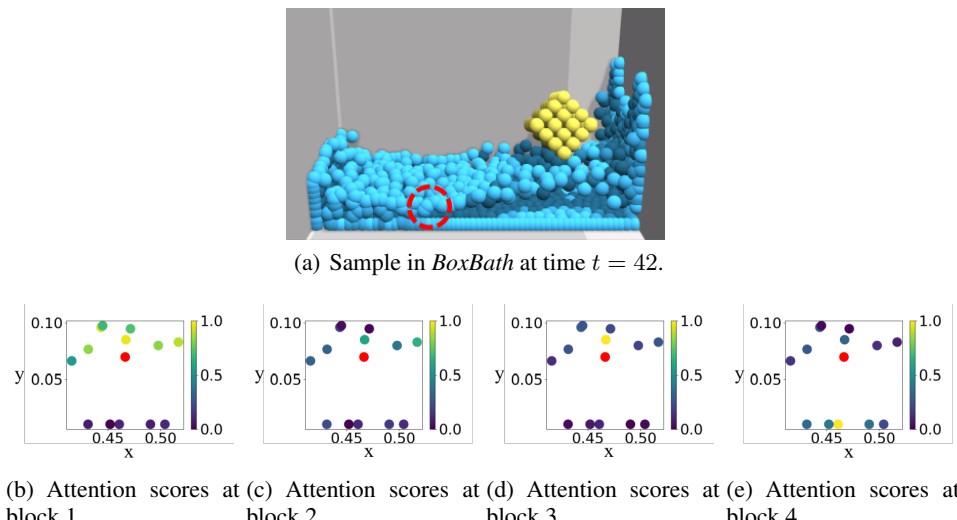

(a) Sample in *BoxBath* at time $t = 42$.

(b) Attention scores at block 1.  (c) Attention scores at block 2.  (d) Attention scores at block 3.  (e) Attention scores at block 4.

Figure 7: (a) is a sample frame on *BoxBath*. We select one particle in the center of red circle to show its attention scores on interactions from neighbors. (b)(c)(d)(e) show the heat maps from uniform attention in each block of SiT. The attention scores on interactions are assigned to the corresponding neighbors, which are shown by different colors. The darker the color, the lower attention score on particles.

ticles with 151 frames for each rollout. There are 2700 rollouts in training set and 300 rollouts in validation set.

**Results**. We display some rollouts for all experiments in this section. The results on *RiceGrip* for CConv are not available because CConv was designed specifically for fluid dynamics. The results on *BoxBath* for CConv are only used as reference of the simulation on fluid parts. And we compare CConv with SiT mainly on the fluid simulations.

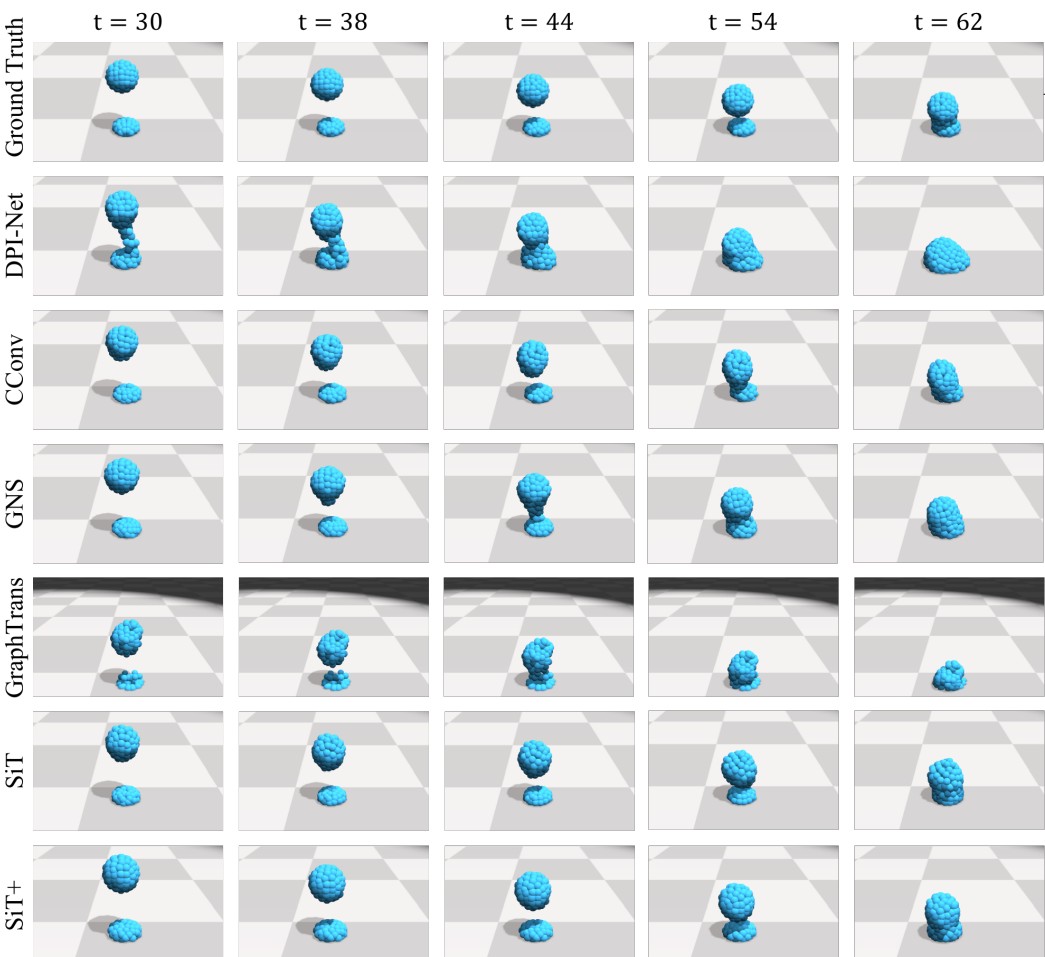

Figure 8: Rollouts on *FluidFall*. Here we display the settings where the larger drop is above the smaller drop. DPI-Net and GNS are likely to mix the drops together, while CConv alters the shape of the upper drop before falling on the floor. GraphTrans has worse performance of maintaining the shape of water drops. SiT changes a little for the shape of the upper drop. When using abstract tokens, which is marked by +, SiT can achieve better results which are closer to the ground truth.

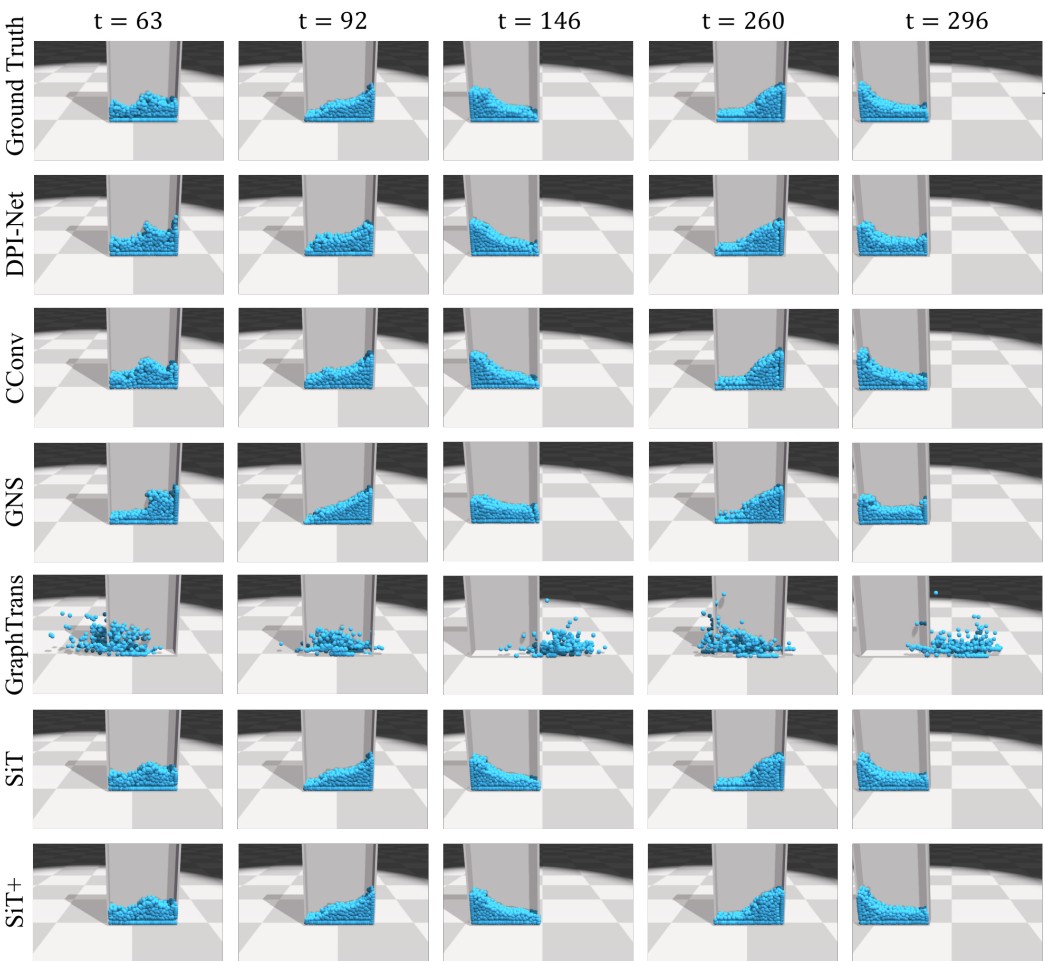

Figure 9: Rollouts on *FluidShake*. DPI-Net still has some artifacts in predicting the wave of water, such as the rollout at $t = 63$. CConv, which is designed specifically for fluid dynamics, rollouts fairly good results, such as the surface of water, in this domain. GNS has artifacts when predicting the waves ($t = 63$) and predicts overly smooth surfaces of fluid ($t = 90$). GraphTrans fails in this domain, suggesting the insufficiency of capturing the complex interactions from outer forces. SiT can also achieve compatible results with less parameters.

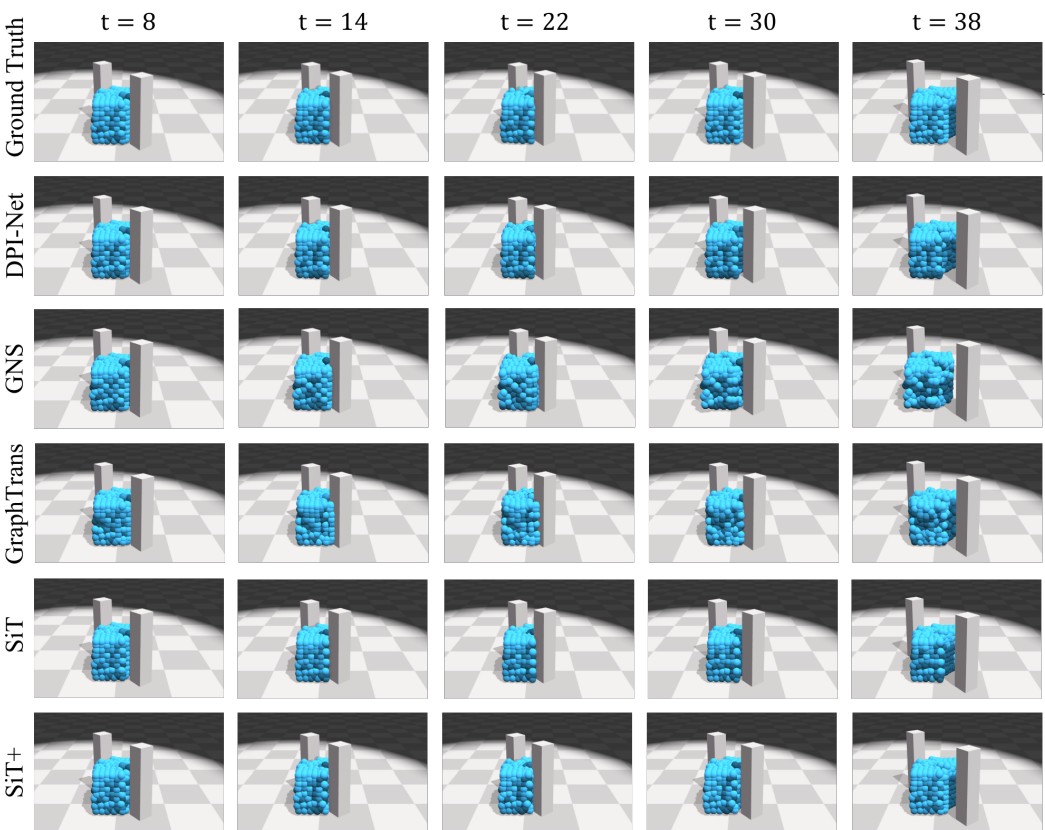

Figure 10: Rollouts on *RiceGrip*. For the surface of the rice, we can see at $t = 38$ that DPI-Net seems to put more pressure to the rice; GNS and GraphTrans has difficulties in maintaining the smooth surfaces of the rice. SiT is capable of predicting faithful rollouts , while SiT with abstract tokens can better simulate the deformable object.

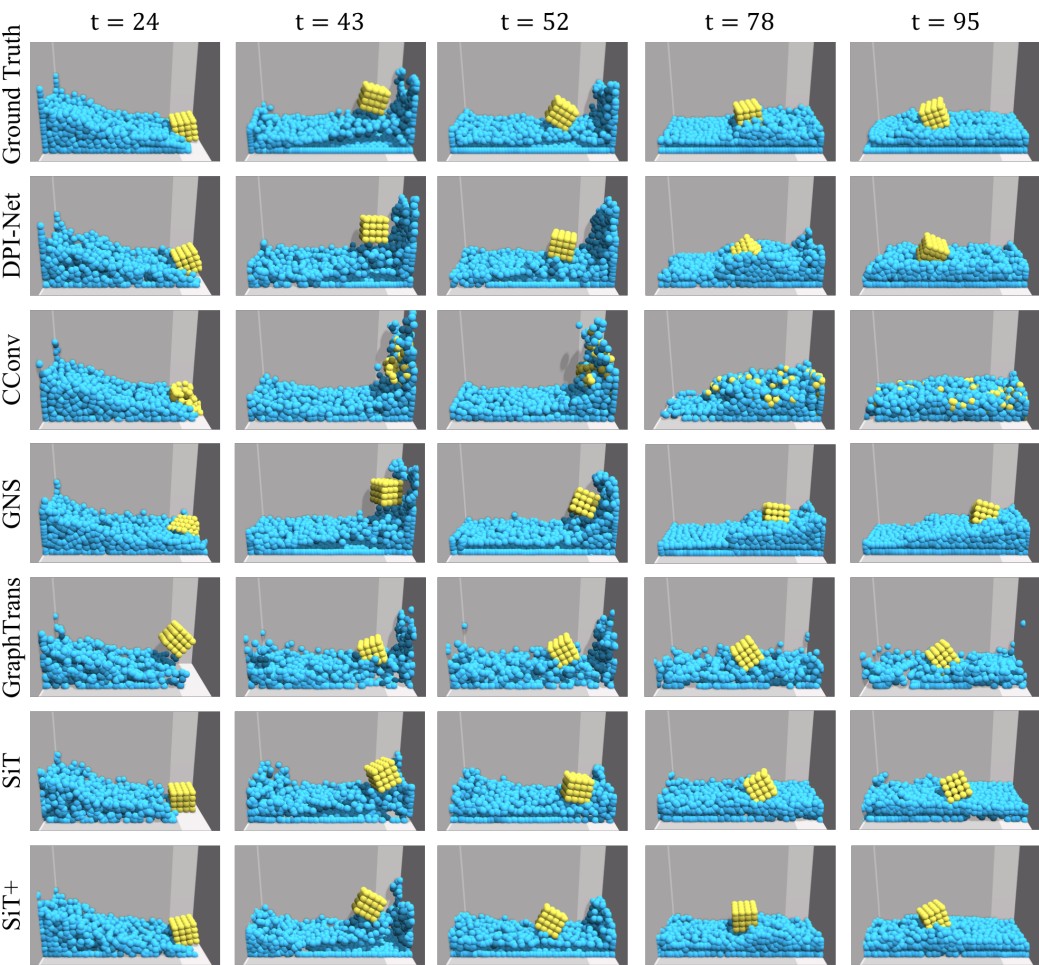

Figure 11:  Rollouts on *BoxBath*. As CConv is designed for fluid dynamics, it treats all particles as fluid and fails in this domain. GNS is able to handle the rotations of rigid box, but has difficulties in modeling the motions of rigid box when pushed by the waves of fluid, suggesting the insufficient abilities of dealing with interactions between different materials. SiT with abstract tokens achieves more faithful results in terms of the rotation of rigid cubics, the interactions between fluid particles.

Table 5: MSE and WMSE results from all models on generalizations. The lists of numbers in *FluidShake* and *RiceGrip* are the range of particles, while the tuples in *BoxBath* denotes number of fluid particles, number of rigid particles, and shape of rigid objects respectively. Training settings are marked by *.

| Methods | FluidShake [450,627]* | | | RiceGrip [570,980]* | |
|---|---|---|---|---|---|
| | [720,900] | [924,1080] | [1104,1368] | [1060,1345] | [1347,1640] |
| DPI-Net | 2.14±0.36 | 2.78±0.58 | 3.53±0.71 | 2.33±30.64 | 0.87±4.00 |
| CConv | 1.86±0.42 | 2.23±0.72 | 2.50±0.77 | N/A | N/A |
| GNS | 2.06±0.75 | 2.82±1.93 | 2.98±1.22 | 0.56±0.31 | 0.57±0.29 |
| GraphTrans | 10.24±3.58 | 12.27±4.80 | 12.72±3.75 | 0.63±0.28 | 0.68±0.37 |
| SiT+ (Ours) | **1.54±0.37** | **1.90±0.63** | **2.30±0.69** | **0.14±0.10** | **0.17±0.15** |

| Methods | BoxBath (960,64,cubic)* | | | | |
|---|---|---|---|---|---|
| | (1280,64,cubic) | (960,41,bunny) | (960,125,cubic) | (960,136,ball) | (960,120,cuboid) |
| DPI-Net | 4.28±0.11 | 2.49±0.23 | 2.75±0.42 | 2.71±0.35 | 2.97±0.38 |
| CConv | 4.30±1.73 | 3.03±0.18 | 3.21±0.82 | 3.80±0.22 | 3.05±1.55 |
| GNS | 2.87±0.19 | 2.22±0.28 | 1.97±0.28 | 2.08±0.22 | **1.81±0.18** |
| GraphTrans | 3.87±0.08 | 3.55±0.07 | 3.92±0.15 | 3.86±0.17 | 3.56±0.13 |
| SiT+ (Ours) | **2.64±0.11** | **1.72±0.08** | **1.74±0.14** | **1.67±0.13** | 1.92±0.32 |

| Methods | BoxBath WMSE (960,64,cubic)* | | | | |
|---|---|---|---|---|---|
| | (1280,64,cubic) | (960,41,bunny) | (960,125,cubic) | (960,136,ball) | (960,120,cuboid) |
| DPI-Net | 3.14±0.39 | 3.65±1.72 | 3.42±0.90 | 3.13±0.68 | 3.96±1.39 |
| CConv | 4.30±1.09 | 3.59±0.51 | 4.09±0.62 | 4.43±0.35 | 3.43±0.90 |
| GNS | 4.55±1.53 | 8.09±3.32 | 3.40±1.06 | 3.60±0.74 | **2.18±0.66** |
| GraphTrans | 3.37±0.75 | 3.59±0.76 | 5.31±0.73 | 5.38±0.69 | 3.81±0.75 |
| SiT+ (Ours) | **1.82±0.21** | **1.81±0.64** | **1.94±0.51** | **1.76±0.35** | 2.86±1.41 |

Table 6: Details of generalization settings.

| Environments | Number of training particles | Number of generalization particles |
|---|---|---|
| FluidShake Large | [450, 720] | [720, 1500] |
| RiceGrip Large | [558, 1060] | [1060, 1798] |
| BoxBath Large Fluid | Fluid: 960. Rigid: 64 | Fluid: 1280. Rigid 64 |
| BoxBath Bunny | Fluid: 960. Rigid: 64 | Fluid: 960. Rigid: 41 |
| BoxBath Large Cubic | Fluid: 960. Rigid: 64 | Fluid: 960. Rigid: 125 |
| BoxBath Large Ball | Fluid: 960. Rigid: 64 | Fluid: 960. Rigid: 136 |
| BoxBath Large Cuboid | Fluid: 960. Rigid: 64 | Fluid: 960. Rigid: 120 |

### 6.2.3 GENERALIZATION DETAILS

We report WMSE results on *BoxBath* and other models' generalization performance for references in Table 5. SiT with abstract tokens achieves reasonable and superior performance.

We release the details of generalization settings in Table 6. For the generalizations of *FluidShake* and *RiceGrip*, we display the range of particle numbers. For the generalizations of *BoxBath*, we change the shape of rigid box into bunny, ball, cuboid, and larger cubic. Specifically, the generalized *RiceGrip* contains particles from 1060 to 1640, while the training set contains particles from 570 to 980; the generalized *FluidShake* contains particles from 720 to 1368, while the training set has 450 to 627 particles; For *BoxBath*, the training set has 960 fluid particles and 64 rigid particles. We first separately increase the fluid particles and rigid particles to 1280 and 125 respectively. Then, we change the shape of rigid object into bunny, ball, and cuboid with particles 41, 136, and 120 respectively. The followings are the rollouts on all environments.

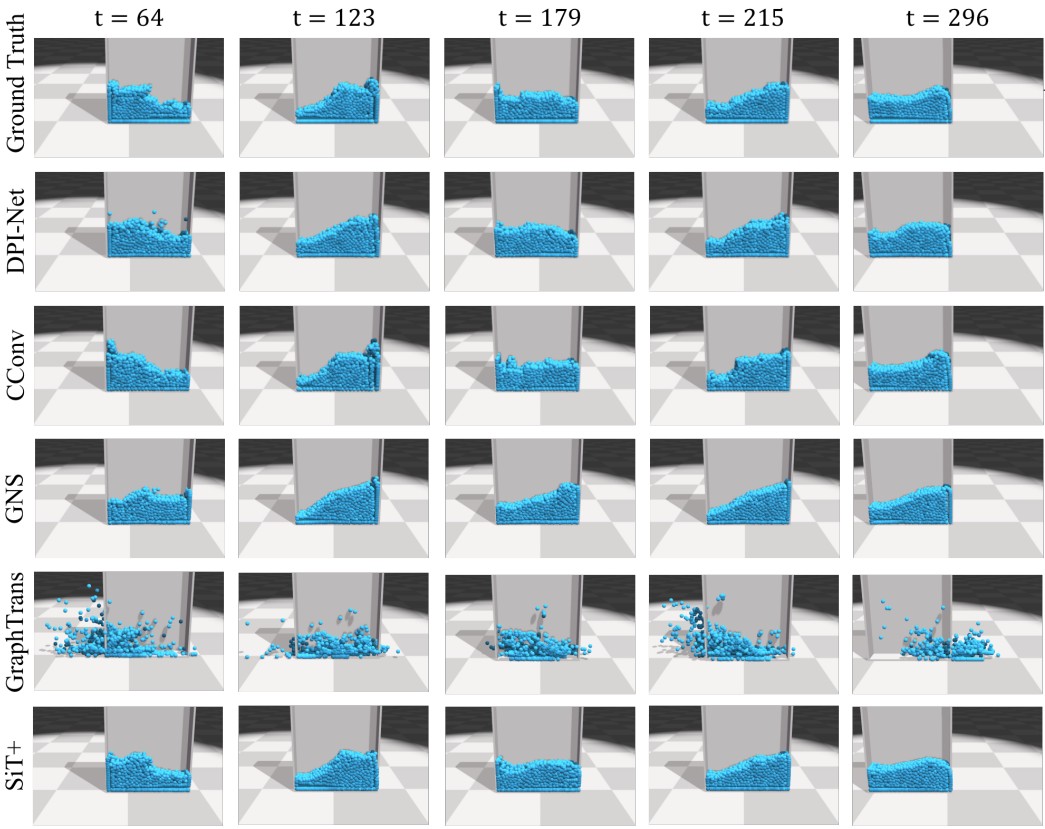

Figure 12: Rollouts on generalized *FluidShake*. DPI-Net achieves fairly good performance, while it still has difficulties to correctly predict the waves, such as the rollout at $t = 123$. CConv and GNS have similar problems, such as the rollout at $t = 215$ and $t = 123$ respectively. GraphTrans still fails in this environment. Our SiT+ can still achieve better performance.

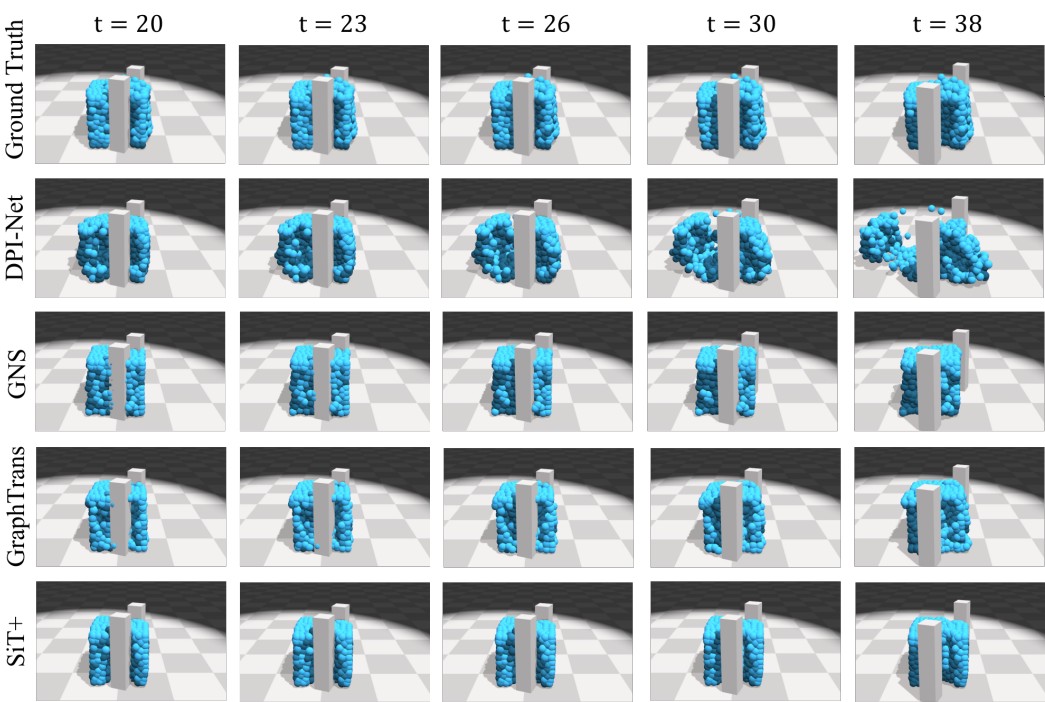

Figure 13: Rollouts on generalized *RiceGrip*. DPI-Net fails and cannot maintain the shape of the deformable object, as is shown in the prediction at $t = 38$. GNS tends to rotate the corner of the rice, as is shown at $t = 38$, the left lower corner is twisted when the grippers do not have contact with the rice's surface. Our SiT+ has better rollouts than GraphTrans, suggesting the effectiveness and robustness of our method.

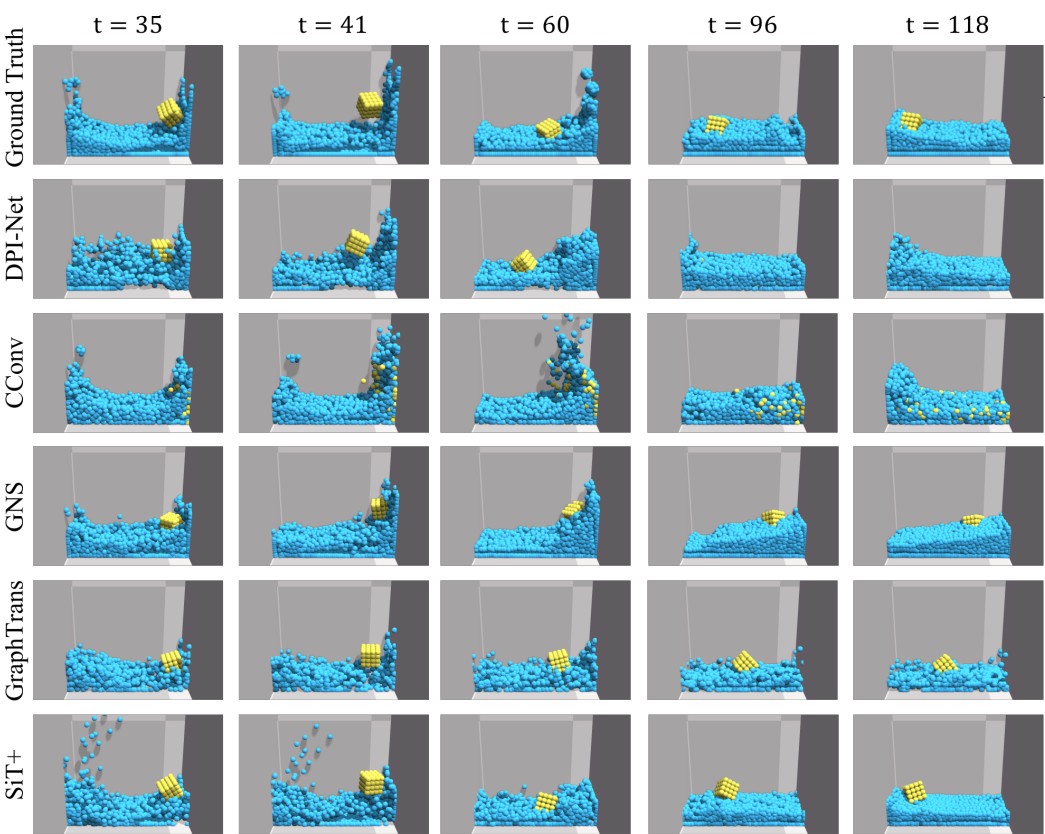

Figure 14: Rollouts on generalized *BoxBath*, where we add more fluid particles. DPI-Net misses the cubic at the end. While GNS is able to simulate the rotations of rigid box, it fails to make further predictions of the rigid after it falls into the water. While SiT+ predicts wrong motions of several fluid particles at the beginning, it still achieve better results compared with other models.

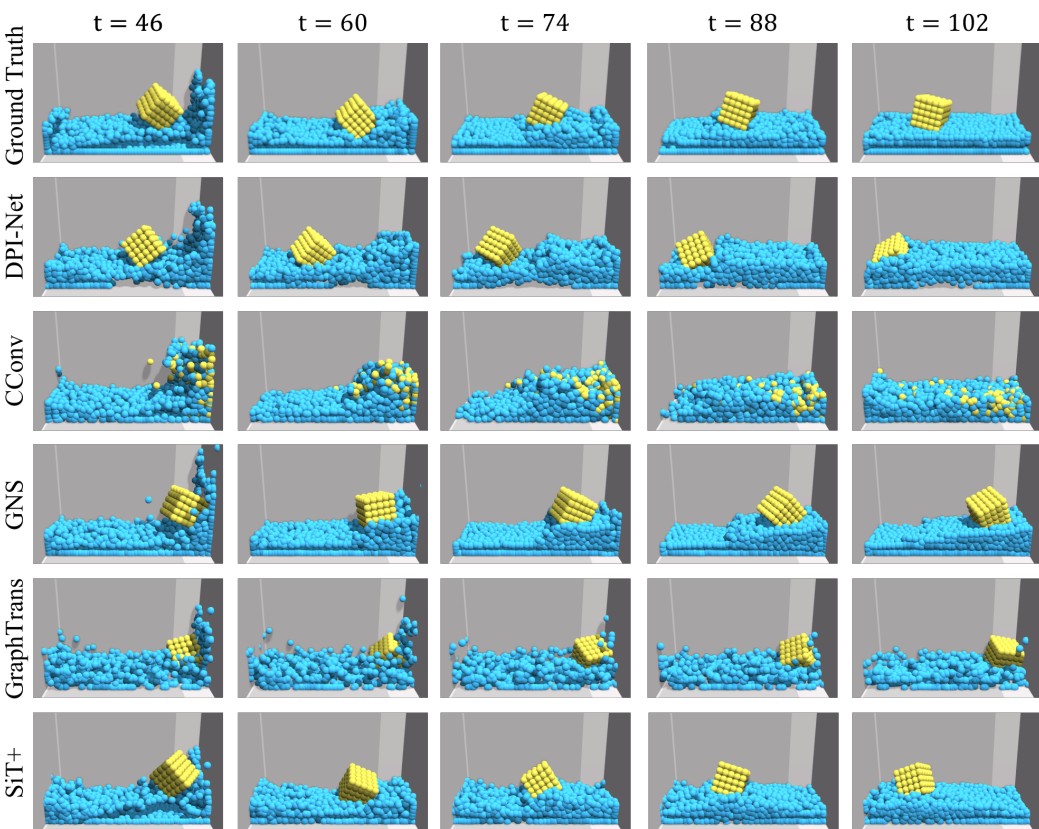

Figure 15: Rollouts on generalized *BoxBath*, where we add more rigid particles. DPI-Net starts to predict wrong rollouts for fluid particles at the beginning, and worse results for the rigid cubic. GNS seems overfit to the rotations of rigid box. SiT+ achieves better results comparing with other models.

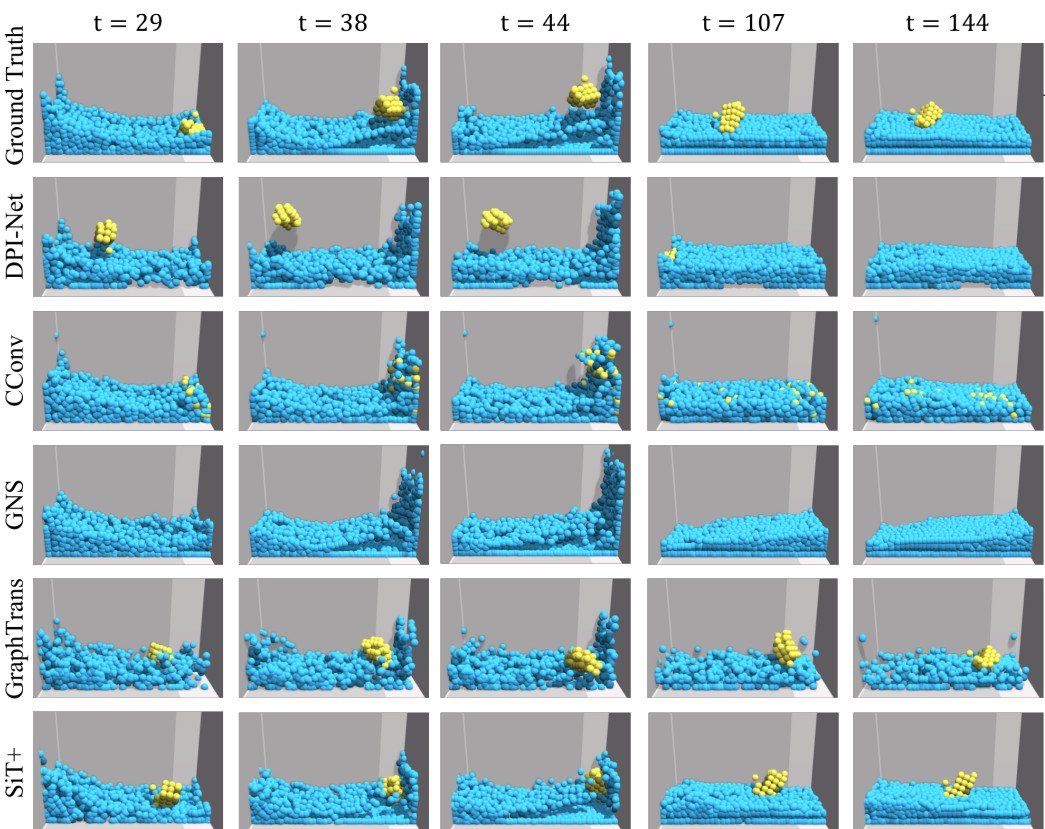

Figure 16: Rollouts on generalized *BoxBath*, where we change the shape from cubic to bunny. The shape of bunny is more challenging, as the ears of the bunny do not have direct connections to the body. While DPI-Net fails in this scene, the bunny is submerged in the rollouts of GNS. SiT+ predicts rollouts as correctly as possible.

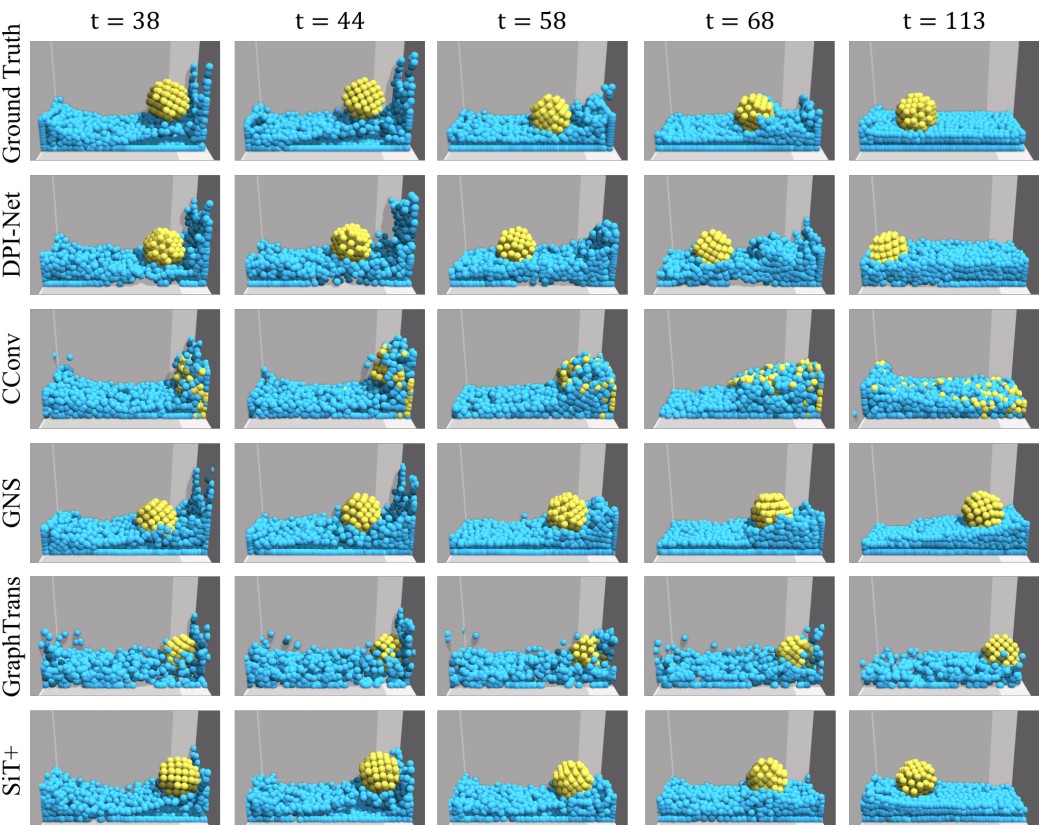

Figure 17: Rollouts on generalized *BoxBath*, where we use a rigid ball. For simple comparison, the positions of the rigid ball suggest models' abilities of simulation. SiT+ predicts more faithful positions of the ball and overall shapes of the fluid.

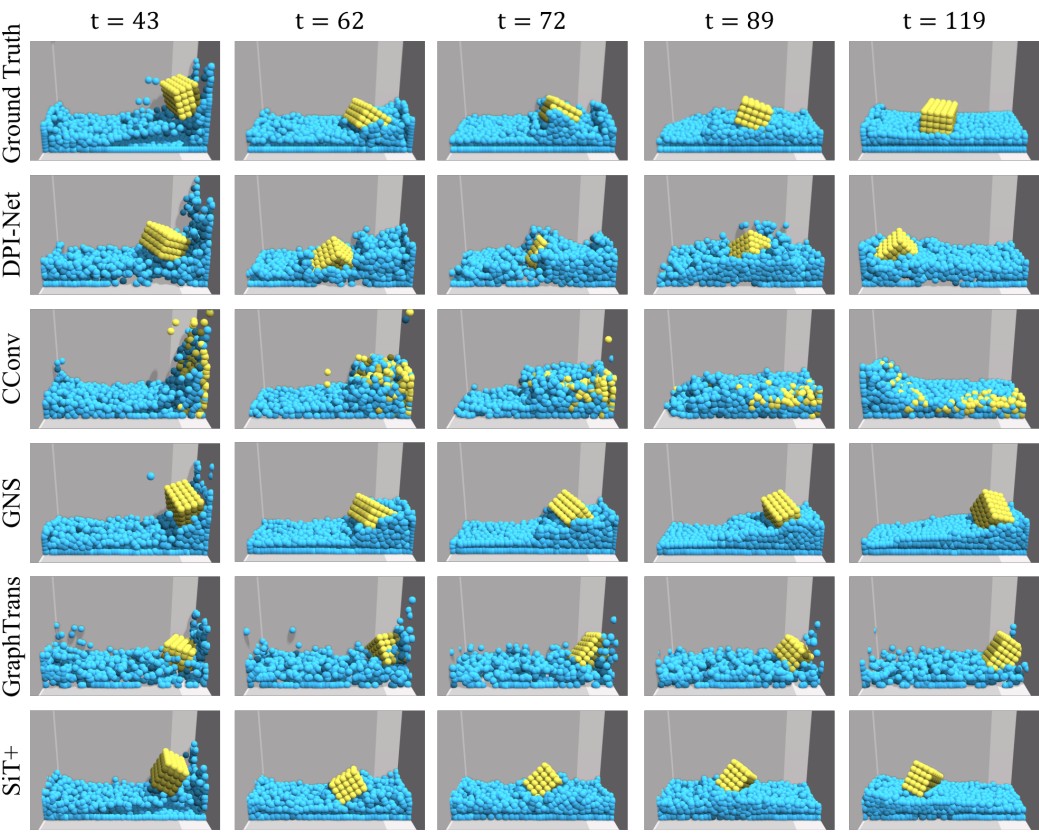

Figure 18: Rollouts on generalized *BoxBath*, where we use a rigid cubiod. It seems that the cuboid is harder to move. DPI-Net fails in this scenes by predicting wrong rotations for the cuboid, GNS overfits to the rotations of rigid box. SiT+ still achieves better results.

