# OpenReview forum: "SiT: Simulation Transformer for Particle-based Physics Simulation"
_ICLR.cc/2022/Conference — ICLR 2022 Submitted_

### Official Review · Reviewer_ZsKn · 2021-10-27

**Correctness:** 2
**Technical Novelty And Significance:** 3
**Empirical Novelty And Significance:** 3
**Recommendation:** 3
**Confidence:** 4

**Main Review:**

Strengths:

S1: Testing the ability of transformers on physical simulation, as opposed to message passing, is indeed an interesting question that needed to be asked. I suspect many groups have probably tried this, and generally found worse performance (Consistent with the findings for the GraphTrans baseline in this paper), but it is good to actually show some published results.

S2: Given that a pure transformer does not seem to work very well, there is another question worth asking: Is there a hybrid between message passing and transformer that could get the best of both worlds?

While I am very supportive of studying those two questions, I am afraid the current experimental setup is not doing a great job to support the claims that are being made due to some weaknesses:

W1: The proposed model, (equation 10) contains alpha and beta terms, and actually the conclusion from the authors is that alpha=0 beta=1 is the best working configuration. Now, if the alpha term is ignored, all that is left is the beta term, and the beta term is mostly a weighted aggregation of equation (8), what the authors call "Interaction Tokens". Looking closely at this function, this is pretty much a high dimensional message function, as used by the GNN baselines (DPI-Net and GNS), the only difference is that in this case this message plays two roles, the role of a high dimensional message, and the role of an edge specific query, which is combined with a receiver node specific key to build an additional attention weight for the aggregation. First of all I would probably not call this a transformer, because the mechanism is neither cross attention, or even self attention, since each edge key is used for only one edge, and never shared for more than one key/query product. Also, the mechanism does not offer the main advantage of transformers which is not having to require to compute specialized edge level embeddings, which are expensive. Instead, it pretty much is a standard message passing model, with an additional scalar attention component.

W2: Now, even if we leave aside whether the model should be called a transformer or not, then the remaining question is: how important is the contribution of the softmaxed "\omega" attention weights in the right term of equation (10). This this is the only major difference between this SiT and the other message passing baselines (e.g. DPI-Net and GNS), the easiest way to study that difference in a clean way would have been to take one of the most competitive message passing baselines (e.g. GNS), and added the proposed attention mechanism (e.g. omega) to the very same implementation, while keeping the rest exactly the same. Still, I tried to gave it my best shot at using evidence presented in the paper to actually find an answer to that question by looking at the comparisons to the closest best performing message passing baseline, but several experimental choices made this very hard (See W3, W4, W5).

W3: It is hard to take much from the comparisons in the BoxBath environments. Specifically, because, as highlighted by comparing the MSE and WMSE losses, the main differences in performance are just depending on whether the model keeps the box together or not. However predictions for the box particles are made with completely different approaches. In DPI-Net, GraphTrans, SiT, and SiT+, a single solid translation and rotation prediction is made for the whole box (it is by design impossible for the box to break apart), while for CConv and GNS the model is asked to produce independent predictions for each particle in the box (so the box can actually break). This difference seems to be the main dominant factor in the comparison between those two groups of models, and specifically between GNS and SiT, however, the explanation of this difference in the text does not seem clear enough. Furthermore, comparison about how to make predictions for the box is completely orthogonal to the topic in the paper, and just makes the results harder to interpret. There does not seem to be a good reason not to use the same approach (either model the box as a whole, or make per particle predictions) for all baselines. Otherwise the results presented in Table 1, and Figure 3 don’t actually tell much about how the extra attention weight contributes to the predictions, and the usage of the WMSE metric seems to just highlight this unfair advantage that some models have over the others when modeling the box. This also seems to directly oppose the main claim at the end of the abstract “Without bells and whistles…achieves superior performance … than existing methods”, as this is precisely a “bell and whistle” that is only used for a subset of the models.


W4: Comparison in FluidFall is also hard to interpret. “though the MSE results are close to each other. SiT can better maintain the shape of drops while DPI-Net and GNS fail due to the wrong prediction of viscosity”. This statement seems to make a quite general claim of why the DPI-Net and GNS fail on this domain, but there is not actual evidence supporting this explanation. First in Figure 3 frames differences are not actually that obvious, but instead we can look at Supplementary Figure 8, where the problem is more apparent. Second, in the animations from the original DPI-Net paper (http://dpi.csail.mit.edu/, “Rollout from our model”), the model predictions for this same domain look qualitatively much better than the results shown on the same domain for DPI-Net. Similarly, the GNS paper also models goop like materials with viscosity (https://sites.google.com/corp/view/learning-to-simulate#h.p_JUaMhhhLYDkV), and this particular artifact that not seem to happen. So not sure if there is any difference in the implementations that could explain this. Third, from my experience, these sort of artifacts appear when (1) the training distribution does not have a lot of variance (like it seems to be the case for this dataset) and (2) a model that has a lot of capacity per message passing layer does not have enough depth to learn effective long range interactions/correlations, so it learns bad short range correlations that exploit the biases in the training dataset and make the model overfit. The reason this happens for DPI-Net and GNS, but not SiT, may be dependent on the choice of hyperparameters so it is unclear, if for a different choice of hyperparameters, results would have told a different story. Note, these biases / failure modes are also more typical of models that are not translation equivariant and try to exploit biases on the absolute positions, but not sure whether the implementations of the models are translation equivariant or not.

W5: GNS is not implemented for FluidShake or Rice Grip. From FluidFall and BoxBath, GNS seems to be the most competitive model with SiT, so it is unclear why no comparison is provided for FluidShake and for RiceGrip. This would probably be ok if it was not because of W3, and W4, but it seems crucially important considering the potential issues with those comparisons.

W6: The role of the abstract tokens in the model is unclear. The results clearly show that SiT+ is better than SiT performance-wise, however, it is unclear why. The authors intent is that they “abstract tokens capture system-agnostic semantics, they can be reused by SiT when generalizing to systems that have same materials but vary in particle amount and configuration.”,  however I don’t think this hypothesis is tested explicitly. Abstract tokens seem to help on FluidFall (which has a single material), and the training distribution on box, batch, for which the whole training datasets consists of 4x4x4 boxes, which cannot be explained by that hypothesis. On RiceGrip, the difference between SiT and SiT+ is larger, however, actually in that domain SiT that worse than DPI-Net, which I find suspicious. On the other hand if I understand abstract tokens correctly, they actually connect to all nodes of the same type, in a bidirectional way, and if this is the case they actually are sort of a global node that allow communication between any two pairs of particles of the same material in two hops, regardless of how far they are. How do we know that the differences observed are not just due to this extra range of communication? To test this, an ablation where “dummy abstract nodes” without the material embedding but with the same connectivity are added should be included. Authors also say “baselines … may regard the latter as part of the intrinsic material semantics and fail to generalize to systems with the same materials but different particle amounts and configurations.”, but this is not backed by evidence, and in fact several of the baselines in the paper have already shown generalization to different shapes at test time.

W7: (Referring to the baselines) “First, it forces each particle to interact with all its nearby particles without providing a selection mechanism, which leads to computational redundancy and prevents the discovery of inherent patterns of particle interaction.”
“They require the particles to interact with all nearby neighbors without selections, which will aggregate redundant and irrelevant neighbors”. This is slightly misleading. Currently, all methods in the paper are using a fixed radius of connectivity to select a number of candidate neighbors (if this claim is True, this should not be necessary for SiT). Second, the SiT model still has to consider all interactions within the radius of connectivity, and learn to discard some by downweighting them. This is something that the baselines could also learn to do, by just outputting messages that are 0 for unnecessary interactions, and a similar plot like Sup. Fig. 7 (b) could be made by showing the norm of the messages. It is true that the explicit scalar weight is a nicer way to disable full edges, rather than disabling all element wise message components, but for the other reasons indicated above, there is no evidence showing that SiT is actually empirically better than the baselines at doing this.


Other observations comments:

O1: I would recommend using the more general term GNN (Graph Neural Network) instead of GCN (Graph Convolutional Network) to refer to the corresponding baselines, as most readers will think of node update rules as done “Kipf et. al” model when reading the term GCN.

O2: The transformer description (eqs. 2, 3, 4), seems to be missing something, as none of these equations that are part of the iterative update of the nodes actually contain learnable weights. I think the computation of separate keys, queries, and values may be missing. Not sure if this is a bug in the math, or if it is actually implemented that way. Perhaps this is also missing in some of the equations (10, 11, 12, 13, 14).

O3: More explanation is needed about specifics of the model, for example, does the encoder uses absolute positions, or does it (like several or the baselines) use relative positions in the encoder, to become translation equivariant? There is also some explanation in the supplementation about number of blocks, number of MLP layers etc, but this is ambiguous and the main hyperparameters (such as number of message passing steps), should be more prominent, and probably in the main text. Also some explanation about how the model and baselines were fine tunes should be included.

O4: Not sure if the definition of WMSE is correct (eq 16). My understanding is that the goal is is to increase the relative weight of particles with materials that appear less often when there is an imbalance of number of particles of different materials. Currently, if you replace equation (15), into equation (16), if is unclear what happens with the N in equation (15). If it gets replaced by an N_k, then it would cancel out with the N_k in equation (16), which means no rebalancing happens, since K is just a constant (number of material types). If it remains as N, then you get N_k/N and the weighting goes into the opposite direction, materials with less particles, get even less weight on the sum on average. It would be useful to just fully expand equation (15).

O5: Not sure if there is something wrong with the top row of figure 3, since the camera position seems to be slightly different for each frame.

O6: “Notice that CConv is not suitable for rigid box, we only report result on BoxBath for reference of simulation on fluid parts.” Why?

O7: In the related work it may be worth mentioning “CConv” before “GNS” as it was published earlier.


**Summary Of The Paper:**

The paper studies the use of transformers for particle-based physics simulation. The main contributions are as follows.

C1. the paper proposes a specific transformer-inspired form of message passing, which, unlike transformer, still does explicit message computation.

C2. the paper proposes a very specific way to encode material type information as abstract tokens.

C3. the paper compares to several non-transfomer based baselines, as well as a pure transformer.

C4. comparisons are performed on a subset of 4 environments including some generalization settings. As presented in the paper, the proposed model seems to perform better than the baselines in most settings.


**Summary Of The Review:**

The intent of the paper is good, and the results could bring interesting insights about whether attention inspired architectures could bring to the simulation domain. However the empirical evaluation makes it very hard to know which improvements are a result of the specific technical additions added to the model (extra scalar attention for the explicit messages and abstract tokens), and which improvements are just due to other small, but possibly very relevant, differences (including different input encoders/decoders, different featurizations, different outputs, etc.) between SiT and the baselines. My recommendation would be to perform the same investigations by adding the extra attention to the existing baselines one by one (or at least to the most promising ones) while keeping other parameters identical and test if the form or proposed attention helps universally. Similarly, the idea of adding extra abstract nodes for material types, and its impact on generalization, could be applied to existing baselines too and tested and studied orthogonally to the attention contribution. This would make the paper conclusions much more insightful and useful than in their current form.

EDIT POST REBUTTAL: I want to thank the authors for the additional experiments and extensive replies. While many interesting ideas are introduced in the paper, the way the comparisons with baselines is performed makes it very hard to really understand which aspects of the proposed architectures is making the model perform better than baselines on some settings.

Since the authors are proposing a new neural network layer, it would be more useful if, rather than compare to different baselines each with their original hyper-parameters from their papers, etc, the authors took the approach of integrating the different parts of the newly proposed layer into existing architectures (possibly including non-simulation settings), and try to understand better that way how the new layer may help in a more apples-to-apples comparison.

---

> ### Author Response · Authors · 2021-11-23
> **Author Response to Reviewer ZsKn [4/4]**
>
> ## Q14: Incorrest order of related work
>
> A:
> As suggested, we correct the order of "CConv" and "GNS".
>
> ## Breif List of Revisions
> ### Common Modification
> 1. Replace "system" by "domain" to denote a specific configuration of data, e.g. FluidFall and FluidShake are two different domains.
> 2. Replace "GCN" by "GNN".
>
> ### Abstract
> 1. Limit the methods discussed in our paper to learning-based methods.
> 2. Add explanations for 'material semantics', 'particle tokens', and 'interaction tokens'.
>
> ### Introduction
> 1. Add explanations for notions, such as 'traditional simulators', 'particle interactions', 'material semantics', 'interaction tokens'.
> 2. Clearer descriptions of our methods.
>
> ### Related Work
> 1. Correct the order of 'CConv' and 'GNS'.
> 2. Add reference of 'COPINGNet'.
>
> ### Methodology
> 1. Add explanations for notions, such as 'domain-specific semantics', 'domain-agnostic semantics'.
> 2. Clearer descriptions of our methods.
> 3. Correct the formulation of WMSE in Equation 16.
>
> ### Experiments
> 1. Update GNS related performances and analysis.
> 2. Add 'sensitiveness of SiT to radius $R$' and 'effectiveness of abstract tokens' into ablation studies.
> 3. Add results for DPI-Net on few shorts learning.
>
> ### Appendix
> 1. Update GNS related performances and analysis.
> 2. Add details about implementations in Appendix 6.1.3.

---

> ### Author Response · Authors · 2021-11-23
> **Author Response to Reviewer ZsKn [3/4]**
>
> ## Q6: Role of abstract tokens is unclear, need compare with dummy nodes; Baselines show generalization to different shapes at test time.
>
> A:
> As suggested, we conduct experiments with "dummy abstract nodes" on both BoxBath and RiceGrip. We randomly initialize the dummy abstract tokens following the normal distribution and fix their value during training. Thus, the dummy abstract tokens do not learn semantic information and the graph has the same connectivities in SiT +. The results are as follows:
>
> | Methods \ MSE(1e-2) | RiceGrip           | BoxBath             |
> | ------------------- | ------------------ | ------------------- |
> | SiT + dummy nodes   | $2.12\pm0.46$      | $3.98\pm0.09$       |
> | SiT + abstract toekns | $0.07\pm0.07$    | $1.57\pm0.06$       |
>
> SiT with dummy nodes achieves worse performances on both of the domains, especially on RiceGrip, where the interactions among deformable particles are more complex. The results suggest that abstract tokens can provide material semantics to improve the performances of SiT.
>
> Secondly, the performances on generalization for baselines are not well, as is shown in the following:
>
> | Methods   | FluidShake [1104, 1368] | RiceGrip [1060, 1345] | BoxBath (960, 136, ball)  |
> | --------- | ----------------- | ------------------ | ------------------ |
> | DPI-Net   | $3.53\pm0.71$     | $2.33\pm30.64$     | $2.71\pm0.35$      |
> | SiT +     | $2.30\pm0.69$     | $0.14\pm0.10$      | $1.67\pm0.13$      |
>
> The results are part of Table 2 in Section 4.2. The baselines indeed can be applied to more complex domains, but they achieve worse results comparing with SiT +.
>
> ## Q7: SiT also use R to interact with others. Baseline can achieve similar selective mechanism
>
> A:
> It's worth noting that we didn't regard the usage of $R$ as a limitation of GCN-based methods.
> In the second paragraph of Section 1, we emphasize that the limitation of existing GCN-based methods is that they does not provide a selective mechanism, where they treat all interactions equally by directly summing the edge features for further processing. In contrast, SiT resort to the self-attention modules to select important interactions. Moreover, SiT also utilizes abstract tokens to capture essential material-aware semantics. The selective mechanism enabled by the self-attention modules and the abstract tokens not only improve the performance of SiT, but also make SiT more robust on different domains.
>
> Second, it is more difficult for baselines to dynamically discard unnecessary interactions. As discussed in Q2, we conduct an experiment where we apply our selective mechanism in GNS and report the quantitative results on FluidShake.
>
> | Methods \ FluidShake  | MSE(1e-2)          | Parameters         |
> | --------------------- | ------------------ | ------------------ |
> | GNS                  | $1.45\pm0.55$      | 1.59M              |
> | GNS + attention       | $1.43\pm0.39$      | 1.59M              |
> | SiT                   | $1.08\pm0.36$      | 0.77M              |
>
> As shown by the results, although with a selective mechanism, GNS achieves lower MSEs and smaller standard deviations, SiT still achieves superior results with less parameters, indicating the effectiveness of our selective mechanism. Intuitively, the self-attention module as well as the interaction tokens provide more room for SiT to select out important interactions.
>
> ## Q8: Use GNN instead of GCN
>
> A:
> As suggested, we replace 'GCN' by 'GNN'.
>
> ## Q9: The computation of separate keys, queries, and values may be missing in equations (2,3,4,10,11,12,13,14)
>
> A;
> For current settings, we apply self-attention in our attention module, where the queries, keys and values are the tokens themselves. And we implement our model that way.
>
> For multi-head attention, which is included in ablation studies and Appendix 6.1.2, we apply learnable weights $Q^{m,l}, K^{m,l}, V^{m,l}$ for queries, keys and values.
>
> ## Q10: Details of the model
>
> A:
> We add more details in Appendix 6.1.3.
> The fine-tuning details for SiT and baselines are in Appendix 6.1.3 "Training" and "Baseline details" respectively.
> We display all the hyper-parameters for SiT in Appendix 6.1.3. Codes will be released to ensure reproducibility.
>
> ## Q11: Wrong definition of WMSE
>
> A:
> As suggested, we correst Equation 16 in Section 3.4.
>
> ## Q12: Misalignment of FluidFall in Figure 3.
>
> A:
> We aligned the rollouts on FluidFall in Figure 3.
>
> ## Q13: Why CConv is not suitable for rigid box.
>
> A:
> As dicussed Q3 and Section 4.1, CConv designs convolutional layers carefully tailored to modeling fluid dynamics and cannot generalize to rigid objects. Admittedly, we can force CConv to learn on BoxBath the same way as mentioned in original GNS paper, but it does not make sense because the CConv is designed for fluid particles. The quantitative results for CConv on BoxBath are only a kind of reference of the simulation on fluid parts. We include CConv as our baseline only to compare the simulation of fluid particles.

---

> > ### Comment · Reviewer_ZsKn · 2021-11-25
> > **Reply to rebuttal.**
> >
> > > As suggested, we conduct experiments with "dummy abstract nodes" on both BoxBath and RiceGrip. We randomly initialize the dummy abstract tokens following the normal distribution and fix their value during training. Thus, the dummy abstract tokens do not learn semantic information and the graph has the same connectivities in SiT +. The results are as follows:
> >
> > Two questions about this experiment. Why initialize to random for the dummy nodes, instead of zeros? Assuming random, why fix for the whole training rather than (this would just make the model overfit more)? For this ablation, when using the abstract nodes, was also the node type then given to the model in any other way (such as give it as a node feature), which is what the baselines do?
> >
> > By suggesting this ablation, I was hoping to see performance go to somewhere between SiT and SiT+, as this model would sit in between, however, the fact that performance gets significantly worse than regular SiT for not apparent reason, indicated this experiment is not really studying the effect of long range message passing via the extra nodes. To clarify again, the idea of this ablation is to run a model with the same connectivity pattern as SiT+ (with some zero-features for the extra nodes added), but for which the node type is given as a node feature, just like in the baselines. Could the authors clarify if this is what was done, and provide any insight of why performance becomes so much worse?

---

> > > ### Author Response · Authors · 2021-11-28
> > > **Author Response to Reviewer ZsKn**
> > >
> > > ## Q1: Why initialize to random for the dummy nodes? Assuming random, why fix for the whole training? Was also the node type then given to the model?
> > > A:
> > > First, since the inputs of particle states are normalized to normal distribution, we also initialized the dummy nodes' following normal distribution to avoid covariate shift.
> > >
> > > Second, the dummy nodes' values are fixed, so they are not updated during backpropagation and do not learn any possible semantics. They are constant inputs denoting the extra dummy nodes.
> > >
> > > Third, since we do not give the node types to abstract tokens, the dummy nodes features did not contain node types for fair comparison in previous experiments.
> > >
> > > Finally, as suggested, we conduct another experiments: we assign constant zero-features for dummy nodes and concatenate the node types in the same way as baselines. The results are as follows:
> > >
> > > | Methods \ MSE(1e-2)   | RiceGrip         | BoxBath             |
> > > | --------------------- | ---------------- | ------------------- |
> > > | SiT                   | $0.15\pm0.12$    | $1.74\pm0.08$       |
> > > | SiT + dummy nodes     | $0.09\pm0.08$    | $2.00\pm0.15$       |
> > > | SiT + abstract toekns | $0.07\pm0.07$    | $1.57\pm0.06$       |
> > >
> > > On RiceGrip, the extra connectivities introduced by abstract tokens indeed bring some benefits and the semantics captured by abstract nodes further improve the performances. On BoxBath, simply adding more connections does not improve a lot, suggesting that the material-aware semantics play more important roles in multi-material interactions.

---

> > > > ### Comment · Reviewer_ZsKn · 2021-11-28
> > > > **Thank you for the extra experiment!**
> > > >
> > > > Thank you for the last experiment, I really appreciate that you took the time to run this.
> > > >
> > > > The results on RiceGrip are what I was expecting, although I guess for rice grip the difference between SiT + dummy nodes and SiT + abstract tokens seem to be well within the error bar, indicating that most of the effect is due to the extra connectivity.
> > > >
> > > > > On BoxBath, simply adding more connections does not improve a lot, suggesting that the material-aware semantics play more important roles in multi-material interactions.
> > > >
> > > > It is not only that it does not improve a lot, but actually it because worse, which is also very surprising, since this model really sits in between the two models, but performance seems worse than either of the two!

---

> > > > > ### Author Response · Authors · 2021-11-30
> > > > > **Author Response to Reviewer ZsKn**
> > > > >
> > > > > ## Q1: SiT with dummy nodes performs worse on BoxBath
> > > > > We list the MSEs for rigid and fluid separately on BoxBath.
> > > > >
> > > > > | Methods\BoxBath       | Rigid MSEs       | Fluid MSEs          |
> > > > > | --------------------- | ---------------- | ------------------- |
> > > > > | SiT                   | $1.06\pm0.56$    | $1.79\pm0.07$       |
> > > > > | SiT + dummy nodes     | $0.69\pm0.35$    | $2.09\pm0.15$       |
> > > > > | SiT + abstract toekns | $1.18\pm0.62$    | $1.59\pm0.06$       |
> > > > >
> > > > > SiT with dummy nodes focuses more on rigid dynamics and achieves larger MSEs on fluid dynamics, leading to larger MSEs in total. Since the remaining time is limited, the results are the best we can get. The additional connections may indeed benefit SiT to some extent.

---

> ### Author Response · Authors · 2021-11-23
> **Author Response to Reviewer ZsKn [2/4]**
>
> ## Q4: Comparison in FluidFall is hard to interpret: Differences in Figure 3 is not obvious; Rollouts seem good in original paper, may exist different implementations from original paper; Lack of variance in dataset and inproper hyperparameters; Translation equivariant or not.
>
> A:
> At first, the main difference on FluidFall in Figure 3 occurs between the two drops: DPI-Net nearly merges the two drops while GNS predicts that the two drops already encounter with each other and start to merge. However, the two drops do not touch each other in the ground truth. As the artifacts generated by different models occur at different timestamp, we select the frame at $t=54$ which can show artifacts from all models. We update the explanations why DPI-Net and GNS generate artifacts in Section 4.1: when the two drops get closer but not close enough to touch each other, the baselines without selective mechanism fail to deal with incorrect interactions introduced by newly added neighbors from different drops
>
> Secondly, the animation frames of FluidFall displayed in our paper adopt different settings from original DPI-Net paper. In our case, the larger drop locates above the smaller drop, while the settings in original DPI-Net paper are the opposite. In terms of the implementations, we list the MSEs from both original DPI-Net paper and our implementation in the following.
>
> | Methods \ MSE(1e-2) | FluidFall          | FluidShake         | RiceGrip           | BoxBath             |
> | ------------------- | ------------------ | ------------------ | ------------------ | ------------------- |
> | DPI-Net (Original)  | $0.15\pm0.03$      | $1.89\pm0.36$      | $0.13\pm0.07$      | $2.03\pm0.41$       |
> | DPI-Net (Implement) | $0.12\pm0.06$      | $1.43\pm0.52$      | $0.11\pm0.21$      | $1.91\pm0.08$       |
>
> We also ran the original codes of DPI-Net on our generated dataset, and achieve the same results as our implementations, where we only modified the pipeline and kept others the same as original codes.
>
> Thirdly, the settings for FluidFall contains only two cases: larger drop is above smaller drop or vice versa with the same possibility. And there are only 189 particles on FluidFall. The settings of generating all dataset are the same as those in original DPI-Net paper. It is likely that DPI-Net and GNS are overfitted to FluidFall. In terms of the hyperparameters, we adopt the same settings as the original paper for both DPI-Net and GNS. For example, the number of message passing steps for GNS is 10.
>
> Fourthly, the implementations for baselines are translation equivariant. We adopt the same strategies as mentioned in the original DPI-Net paper: we normalize the data before they are fed into the network. The mean and standard deviation are computed on the whole training dataset separately for each domain.
>
> ## Q5: GNS is not implemented for RiceGrip and FluidShake
>
> A:
> As suggested, we implement GNS on RiceGrip and FluidShake.
> The hyper-parameters for GNS are the same as original paper.
> On FluidShake, the input data only come from the current frame.
> On RiceGrip, the input data concatenate a history of 3 previous frames.
> We report the quantitative results as follows:
>
> | Methods \ MSE(1e-2) | FluidShake         | RiceGrip           |
> | ------------------- | ------------------ | ------------------ |
> | GNS                 | $1.45\pm0.55$      | $0.38\pm0.25$      |
> | DPI-Net             | $1.43\pm0.52$      | $0.11\pm0.21$      |
> | SiT                 | $1.08\pm0.36$      | $0.15\pm0.12$      |
> | SiT +               | $1.08\pm0.39$      | $0.07\pm0.07$      |
>
> The results show that SiT still achieve superior results on these two domains.
> On FluidShake, GNS has difficulties in handling the outer forces with inputs from only current frame.
> On RiceGrip, GNS does not perform well because it dose not have selective mechanism in SiT, leading to the difficulties in modeling complex deformations.

---

> > ### Comment · Reviewer_ZsKn · 2021-11-25
> > **Reply to rebuttal.**
> >
> > > It is likely that DPI-Net and GNS are overfitted to FluidFall. In terms of the hyperparameters, we adopt the same settings as the original paper for both DPI-Net and GNS. For example, the number of message passing steps for GNS is 10.
> >
> > But if the only explanation to this is overfitting then this should be investigated in more detail. If the different models are performing differently mostly because, given the hyperparameters of the original papers, they overfit more than the model specific to this paper, then the first thing to try, should be to adjust the hyperparameter of the existing baselines. If they only reason SiT does better is not because it is a better model, but because it has been tuned to work well on those specific datasets, then the comparison on this non-standarized benchmarks, are not really that meaningful. For example, if I understand correctly, assuming alpha=0 and beta=1 SiT is the same as DPI-Net, except for the attention weights. So if you remove attention from the SiT model in the table, do you actually get the same performance shown DPI-Net in the table? Because if you don't then the current comparisons are just impossible to interpret.

---

> > > ### Author Response · Authors · 2021-11-28
> > > **Author Response to Reviewer ZsKn**
> > >
> > > ## Q1: SiT is better because it is tuned on specific dataset.
> > > A:
> > > First, we focus on the effectiveness of model instead of fine-tuning models on each specific domain to get better results. SiT adopts the same model hyperparamters on all domains except that it has 2 blocks on FluidFall, where DPI-Net also has 2 blocks. We adopt the same training hyperparameters, such as learning rate, as DPI-Net and did not fine-tune on each domain additionally. It is worth noting that DPI-Net would have already explored the impact of hyperparameters on different domains, otherwise DPI-Net would report much better results on FluidFall. As discussed in Section 4.1, the key reason for the artifacts on FluidFall is that without selective mechanism, DPI-Net and GNS  would have difficulty in dealing with the incorrect interactions introduced by newly added neighbors from different droplets when two droplets approach each other but not close enough to touch. Such artifacts also occur when the smaller droplet is above the larger droplet, but this is less obvious.
> > >
> > > Second, FluidFall is a simple domain which could only check whether the model has the basic ability to simulate. On BoxBath, FluidShake, and RiceGrip, which are far more complex than FluidFall, better results with the same dataset settings are more convincing to illustrate the effectiveness of the model.
> > >
> > > Third, as discussed in Q1 of "Reply to rebuttal" for "Author Response to Reviewer ZsKn [1/4]", SiT takes effect thanks to the three-head attention. We design SiT following the style of Transformer, and the only same architecture between SiT and other GNN-based models is the interaction tokens (Equation 9). Other key differences between SiT and DPI-Net are that DPI-Net adopts hierarchy structures and shared-weight networks for instantaneous propagation of forces, which is not the case in SiT.

---

> ### Author Response · Authors · 2021-11-23
> **Author Response to Reviewer ZsKn [1/4]**
>
> We thank reviewer for the valuable feedback. We have revised the paper to reflect the feedback, where revision is marked with magenta color.
> Below we address each concern in detail.
>
> ## Q1: When ignore alpha, the beta term is mostly a weighted aggregation of interaction tokens, each edge key is used for only one edge and never share with other product, not attention; have to compute edge level embeddings, which is not the advantages of transformer.
>
> A:
> The intention of designing interaction tokens is to enable particle tokens to filter out unnecessary interactions according to the key/query product.
>
> Admittedly, when $\alpha=0$, the interaction tokens and their updating formulations share large similarity with edges in GCN-based methods. However, $\alpha$ and $\beta$ provide a flexible way to adjust the modeling of interactions and particles in SiT. While here we only use discrete values ${0,1}$ for $\alpha$ and $\beta$, they can be relaxed to have real values, further increasing the flexibility of SiT.
>
> Secondly, we have conducted ablation studies to explore the importance of "edge level embeddings". Part of results in Table 3 are shown here. It is obvious that the high-dimentional embeddings for interaction tokens can bring large benefits to the model. In other words, we combine the adavantages of GCN-based methods (interaction embeddings) with Transformer (selective mechanism) to design SiT.
>
> | Configurations                      | FluidShake (1e-2) |
> | ----------------------------------- | ----------------- |
> | Transformer w/o edge embedding      | $21.03\pm10.54$   |
> | Transformer w edge embedding (SiT)  | $1.08\pm0.36$     |
>
> ## Q2: How important is the contribution of the softmaxed "$\omega$" attention weights in the right term of equation (10)
>
> A:
> As suggested, we also conduct the experiment where we equip GNS with an attention module thus providing it with a selective mechanism.
> Below we show the results of GNS, GNS+attention, and our SiT on the FluidShake environment. Hyper-parameters are chosen to ensure a fair comparison.
>
> | Methods \ FluidShake  | MSE(1e-2)          | Parameters         |
> | --------------------- | ------------------ | ------------------ |
> | GNS                   | $1.45\pm0.55$      | 1.59M              |
> | GNS + attention       | $1.43\pm0.39$      | 1.59M              |
> | SiT                   | $1.08\pm0.36$      | 0.77M              |
>
> As shown by the results, although with a selective mechanism, GNS achieves lower MSEs and smaller standard deviations, SiT still achieves superior results with less parameters, indicating the effectiveness of our selective mechanism. Intuitively, the self-attention module as well as the interaction tokens provide more room for SiT to select out important interactions.
>
>
> ## Q3: Not apply rigid constraints for GNS and CConv.
>
> A:
> As suggested, we retrain the GNS by the same rigid constraints without any other modifications. It is worth noting that we only adopt the current frame as the source of inputs and do not consider the history of previous frames in our settings, which is different from origianl GNS paper. The hyper-paramters, such as the number of message passing is 10, are the same with the original paper. We report the quantitative results on BoxBath as follows.
>
> | Methods \ BoxBath     | MSE(1e-2)          | WMSE(1e-2)         | Parameters         |
> | --------------------- | ------------------ | ------------------ | ------------------ |
> | DPI-Net               | $1.91\pm0.08$      | $1.52\pm0.33$      | 1.98M              |
> | GNS                   | $1.45\pm0.12$      | $2.86\pm0.83$      | 1.59M              |
> | GNS + constraints     | $1.77\pm0.87$      | $2.62\pm0.87$      | 1.59M              |
> | SiT                   | $1.74\pm0.08$      | $1.42\pm0.29$      | 0.77M              |
> | SiT + abstract tokens | $1.57\pm0.06$      | $1.39\pm0.31$      | 0.77M              |
>
> As is shown, when the baselines are trained with the same rigid constraints, while GNS achieves lower WMSE. SiT still achieves superior performance because of the abstract tokens and selective mechanism. The models will make a compromise between MSE and WMSE when using the rigid constraints. GNS has larger WMSE because it does not have selective mechanism in SiT, making it harder for rigid partilces to deal with interactions from different materials when adding rotation constraints.
>
> Second, as we mentioned in Section 4.1, where we compare CConv with SiT, CConv designs convolutional layers carefully tailored to modeling fluid dynamics and cannot generalize to rigid objects. Admittedly, we can force CConv to learn on BoxBath the same way as mentioned in original GNS paper, but it does not make sense because the CConv is designed for fluid particles. The quantitative results for CConv on BoxBath are only a kind of reference of the simulation on fluid parts. We include CConv as our baseline only to compare the simulation of fluid particles.

---

> > ### Comment · Reviewer_ZsKn · 2021-11-25
> > **Reply to rebuttal.**
> >
> > Thank for the responses, see some follow up below:
> >
> > > provide a flexible way to adjust the modeling of interactions and particles in SiT. While here we only use discrete 0, 1 values  for $\alpha$ and $\beta$, they can be relaxed to have real values, further increasing the flexibility of SiT
> >
> > While I agree this provides flexibility, I am not sure this flexibility brings much value for the paper. If the main conclusion is that disabling the transformer part of the model works best, then to me the conclusion is that the best performing model is not really a transformer, the model name Simulation Transformer (SiT) should not have the word "Transformer" in it, and a different acronym should be use for the actual model in the tables. Otherwise if other papers in the future reference this as a baseline for their tables, it will be very confusing that the model called Simulation Transformer does not actually have a transformer in it.
> >
> > > we have conducted ablation studies to explore the importance of "edge level embeddings"
> >
> > To clarify, I did not say the edge level embeddings are not useful, I am just saying that edge level embeddings are precisely what make this model not a transformer, edge level embeddings are the same as the messages in propagation networks or interaction networks, and they are much more expensive to compute than transformers. So the model really has a lot more in common with propagation networks + attention, than with transformers.
> >
> > > As suggested, we also conduct the experiment where we equip GNS with an attention module thus providing it with a selective mechanism.
> >
> > Mathematically, could the authors clarify what the difference between "GNS + attention", and "SiT (alpha=0, beta=1)" is? My understanding is that once you add attention to GNS, they are pretty much the same model, so I would like to know what the main factor contributing to the gap in performance between GNS + Attention and SiT presented in this table is and why the two models have a different number of parameters, when they are almost equivalent mathematically.
> >
> > > GNS has larger WMSE because it does not have selective mechanism in SiT, making it harder for rigid partilces to deal with interactions from different materials when adding rotation constraints.
> >
> > I am not sure I believe this explanation, because DPI-Net (also in that table), which is also more similar to GNS in some ways, also does not have any selective mechanism, and WMSE for DPI-Net is much closer to SiT.
> >
> > (as a side note the parameters in the GNS model seems to have decreased from 4.77M to 1.59M was the model changed?)

---

> > > ### Author Response · Authors · 2021-11-28
> > > **Author Response to Reviewer ZsKn**
> > >
> > > We thank reviewer for further feedback. Below we address each concern in detail.
> > >
> > > ## Q1: Not a Transformer
> > > A:
> > > It is worth noting that the interaction tokens are bridges for "three-head attention". Let's have a deep look about the equation 9 and equation 14. In the following, we omit activation function and timestamp for simplicity.
> > >
> > > First, we have
> > >
> > > $$
> > > \mu_{ij}^{l+1} = MLP([(v_i^l)^\top, (v_j^l)^\top, (\mu_{ij}^l)^\top]) = W\cdot [(v_i^l)^\top, (v_j^l)^\top, (\mu_{ij}^l)^\top]^\top = W_1 v_i^l + W_2 v_j^l + W_3 \mu_{ij}^l
> > > ,$$
> > >
> > > where vectors are column vectors, $[\cdot]$ denotes concatenation, $W \in \mathbb R^{d_h\times 3d_h}$ is learnable weight for MLP, and we can split $W$ into three parts: $W_i \in \mathbb R^{d_h\times d_h}, i=\{1,2,3\}$. Then we have
> > >
> > > $$
> > > (v_i^l)^\top \mu_{ij}^{l+1} = (v_i^l)^\top (W_1 v_i^l + W_2 v_j^l + W_3 \mu_{ij}^l） = (v_i^l)^\top W_1 v_i^l + (v_i^l)^\top W_2 v_j^l + (v_i^l)^\top W_3 \mu_{ij}^l.
> > > $$
> > >
> > > Such formulation has similar formation comparing with the query/key product in multi-head attention formulation: $(W^Qq)^\top(W^Kk)=q^\top (W^Q)^\top W^K k = q^\top W' k$. Our $v_{i}^l$ acts as the query and $v_i^l, v_j^l, \mu_{ij}^l$ act as the keys. Thus, $(v_i^l)^\top \mu_{ij}^{l+1}$ is a summation of 3 attention weights between query ($v_i^l$) and keys ($v_i^l, v_j^l, \mu_{ij}^{l}$). Therefore, our uniform attention is a special case of Transformer.
> > >
> > > ## Q2: Edge level embeddings make the model not a transformer
> > > A:
> > > As discussed in Q1, the interaction tokens ("edge level embeddings") are the bridges for three-head atttention, which is a special case of multi-head attention. We combine the advantages of GCN-based models (edge level embeddings) with Transformer (attention) to design SiT.
> > >
> > > ## Q3: Main factor contributing to the gap between GNS+Attention and SiT; Why different number of parameters
> > > A:
> > > Firstly, one of the main differences is that GNS concatenates the outputs of attention module with the nodes features before MLP, while SiT directly forwards the attention outputs to the MLP following the style of Transformer. In GNS, such concatenation leads to redundant information and weakens the role of the attention results, because the attention features, which already aggregate necessary interactions and information, are fed into the MLP together with the redundant node features. And it is difficult for the MLP to learn to suppress the node features without any auxiliary. On the other hand, SiT directly forwards the attention features to MLP, which is more efficient and eliminates redundant information, providing more room for SiT to select out important interactions.
> > >
> > > Secondly, our self-attention modules, which we refer to as the dot product and weighted summation, do not introduce any parameters. As discussed above, it is worth noting that SiT applies MLP on the output of the attention module $v_i^{l+1}$, while GNS applies MLP on the concatenation of $v_i^l, v_j^l, \mu_{ij}^l$, leading to more learnable parameters. Moreover, as mentioned in original GNS paper, GNS need 10 message passing steps to capture enough long range dependencies and achieve faithful predictions, while SiT only requires 4 blocks with attention module, which further reduces the parameters in SiT.
> > >
> > > ## Q4: Why GNS has larger WMSE, while DPI-Net has closer WMSE to SiT
> > > A:
> > > The basic block for GNS is the interaction network (IN) blocks, which is disuccsed in origianl DPI-Net paper and "only considers local information and cannot handle instantaneous propagation of forces". Indeed, GNS could better model long range dependencies by stacking more IN blocks to some extend. But DPI-Net adopts improved propagation networks and hierarchy structures to achieve better performances than GNS when using only the current frame to rollout next frame. More importantly, DPI-Net adopts two separate prediction heads for rigid and fluid respectively, which make it easier to learn material-specific semantics, such as the rotations for rigid. Predictions for different materials do not intervene each other. Therefore, the rigid constraints will make a bigger difference for rigid predictions. On the other hand, both GNS and SiT adopts only one prediction head and SiT achieves better results due to the selective mechanism, which can effectively deal with interactions from different materials.
> > >
> > > In addition, there is no changes in the GNS model and it was a mistake to count the GNS's parameters, specifically the parameters in Adam optimizer were also counted in the previous version.

---

> > > > ### Comment · Reviewer_ZsKn · 2021-11-28
> > > > **Thank you for your replies**
> > > >
> > > > > GNS need 10 message passing steps to capture enough long range dependencies and achieve faithful predictions, while SiT only requires 4 blocks with attention module, which further reduces the parameters in SiT.
> > > >
> > > > If I understand correctly, all models use the same connectivity pattern (except for the extra connectivity given by the abstract tokens to SiT+). So I am not sure that statement is correct, since all models that use local connectivity (except for SiT+) and I don't see how SiT could model a long range interaction that is more than 4 hops away in the past.
> > > >
> > > > I don't really know exactly how strong of a claim of using 10 message steps specifically was made in the original GNS paper, and whether 10 mps were specifically necessary for all of the datasets, or was the minimum to make the model work for all datasets in that paper, but it seems that the correct thing to do here for an insightful comparison would be to also use GNS with 4 message passing steps rather than 10 specially if overfitting is observed to be a problem?
> > > >
> > > > > Q1: Not a Transformer
> > > >
> > > > Fair enough, I guess we should not discuss a lot about naming, I would personally find the "Transformer" denomination for this model more confusing that helpful.
> > > >
> > > > > In the following, we omit activation function and timestamp for simplicity. First, we have
> > > >
> > > > The problem is that this oversimplifies things. If you have a DeepMLP with activations, you cannot actually perform that decomposition. It is only possibly to perform that decomposition for the first layer of the MLP before the activation. And if you actually use a single MLP layer without activation, then the resulting model is more similar to a Graph Convolutional Network than an InteractionNetwork.
> > > >
> > > > > such concatenation leads to redundant information and weakens the role of the attention results. And it is difficult for the MLP to learn to suppress the node features without any auxiliary.
> > > >
> > > > I agree that these are possible explanations, however it feels a bit speculative, considering that really side to side comparisons using same model depth and similar number of model parameters are not made.
> > > >
> > > >
> > > > > More importantly, DPI-Net adopts two separate prediction heads for rigid and fluid respectively, which make it easier to learn material-specific semantics, such as the rotations for rigid.
> > > >
> > > > I thought in the last version of the experiment a similar approach was also included for GNS?

---

> > > > > ### Author Response · Authors · 2021-11-30
> > > > > **Author Response to Reviewer ZsKn**
> > > > >
> > > > > We thank reviewer for immediate feedback. Below we address each concern in detail.
> > > > >
> > > > > ## Q1: 4 mps comparisons.
> > > > > At first, we adopt 10 mps for GNS because GNS would be better when applying more mps as mentioned in GNS paper, and 10 mps is the default setting. They reported the results on GOOP, which is a different domain.
> > > > >
> > > > > As suggested, we compare GNS with 4 mps and apply attention to it. The results are as follows:
> > > > >
> > > > > | Methods \ FluidShake  | MSE(1e-2)          | Parameters         |
> > > > > | --------------------- | ------------------ | ------------------ |
> > > > > | GNS                   | $1.38\pm0.94$      | 0.70M              |
> > > > > | GNS + attention       | $1.35\pm0.47$      | 0.70M              |
> > > > > | SiT                   | $1.08\pm0.36$      | 0.77M              |
> > > > >
> > > > > Indeed GNS with 4 mps is slightly better than the one with 10 mps on FluidShake, and the attention modules could also improve its performance. But SiT is still better.
> > > > >
> > > > > ## Q2: "Omit activation function" oversimplifies;
> > > > > First, we adopt one MLP layer in practice. Indeed it is similar to GCN, but SiT resorts to the self-attention modules to select important interactions and updates particle tokens using interaction tokens. Moreover, since we use ReLU as activation function, we can always find a matrix $W'$ satisfying
> > > > > $$
> > > > > W'x = ReLU(Wx)
> > > > > $$
> > > > > by assigning 0 to the coresponding rows of $W'$ for non-activated positions. When it comes to multiple layers, we still can use $W'$ to denote a sequence of dot product of matrix. Therefore, we can always decompose the equation into $(v_i^l)^\top W_1 v_i^l + (v_i^l)^\top W_2 v_j^l + (v_i^l)^\top W_3 \mu_{ij}^l$, which is the format of three-head attention.
> > > > >
> > > > > ## Q3: Same model depth comparisons
> > > > > As discussed in Q1, SiT still achieves better results than GNS, which has the same number of blocks (mps). Therefore, it is possible that such concatenation would introduce redundant information.
> > > > >
> > > > > ## Q4: Does GNS adopt two separate classification head?
> > > > > Only DPI-Net uses separate prediction heads in all our experiments.

---

### Official Review · Reviewer_dYcg · 2021-10-31

**Correctness:** 2
**Technical Novelty And Significance:** 2
**Empirical Novelty And Significance:** 2
**Recommendation:** 3
**Confidence:** 4

**Main Review:**

- In the abstract the sentence 'most existing particle-based simulators adopt graph convolutional networks to model the underlying physics of particles' seems off. For most learning-based methods this might be true, but there of course exists a lot of simulators that are not learning-based.
- In the abstract it is not clear what is meant by 'material semantics', 'particle tokens', 'interaction tokens'.
- S1: 'Different from traditional simulators ...'. Here it would help to provide a few examples and references.
- S1: 'Consequently, they follow the same pipeline ...'. It is not clear which pipeline is meant by 'the same pipeline'. Does it mean the same pipeline is used to model different material properties? Please clarify.
- S1: 'where they can robustly estimate the dynamics of a system with varying number and configuration of particles'. This is a strong claim, please provide a reference or tone-down this statement.
- S1 'rich semantics'. Please define what is meant by this.
- S1 'deploys an abstract token'. Not clear what this means. Please clarify.
- S1 The second to last paragraph about the contributions of the proposed paper remains too vague. The text refers to several concepts (e.g. like interaction tokens, trainable abstract tokens, intrinsic material characteristics, system-specific semantics, good balance between efficiency and complexity, particle interactions etc.) that have all not been introduced up to this point and it is not clear what the contributions of the proposed paper are compared to existing concepts.
- S3.1 'The goal of a simulator is to learn a model ...' Does this refer to a learning-based simulator? In other literature simulators are commonly not described to 'learn a model'. It might be useful to more carefully define what is meant by a simulator in the context of this paper.
- S3.2 'a selective mechanism is thus in great need to help the simulator'. This sentence is not clear. Is this supposed to mean 'a selective mechanism is needed to help the simulator'?
- 'is naturally a good backbone of building'. Why? It would help to add a few more details here. Are transformers 'good' because particles can be interpreted as discrete tokens?
- S3.2 'considering all possible interactions is computationally redundant' This is not clear. Why would that be the case?
- S3.3 'rich semantics': up to this point the text has mentioned the term 'semantics' several times without clearly defining what is meant by it.
- S3.3 'the semantics of a system is scattered in the tokens'. Without a clear definition of what is meant by 'semantics', sentences like these are difficult to comprehend.
- S3.3 'To improve generalization ability of SiT and disentangle system-specific semantics from its system-agnostic counterparts'. This sentence is not clear. What is meant by 'disentangle system-specific semantics'? What is meant by 'system-agnostic counterparts'? Please provide more details.
- S3.3 'Once abstract tokens capture system-agnostic semantics'? At this point 'system semantics' has not be defined, therefore this sentence is not clear.

W.r.t the related work, I consider the following recent work on simulating rod-dynamics with GNNs missing:

H. Shao, T. Kugelstadt, W. Pałubicki, J. Bender, S. Pirk, D. L. Michels, Accurately Solving Rod Dynamics with Graph Learning, Conference on Neural Information Processing Systems (NeurIPS), 2021

**Summary Of The Paper:**

This paper proposes to learn particle dynamics of physical systems by the means of a transformer. In particular, the paper investigates using a vanilla transformer as well as a more customized variant called 'Simulation Transformer'. Both networks are validated based on variety of experiments of simulations of fluids, rigid and deformable objects, as well as through comparisons to existing approaches.

**Summary Of The Review:**


Overall, I am very supportive of the topic this paper addresses and generally also of using transformers for solving physical systems. Using transformers for solving physics is an interesting research direction. However, in its current form I am not enthusiastic about this paper which is due to the following main reason:

(1) While the paper is generally well-written, several sentences are left to vague and unclear. In the introduction several concepts and terms are used without carefully introducing them. This makes understanding the text difficult and blurs what the contribution is. Also, some sentences almost read like a collection of buzzwords which makes is very difficult to comprehend what the actual meaning is. Consequently, while the experiments and results are technically sound, I got the impression more care should be taken in carefully explaining and formalizing the concepts that are aimed to be discussed in the paper.

(2) With 12 pages the Appendix provides a significant amount of additional material, some of which can be considered quite relevant for the main contribution of the paper (e.g. Table 4). This indicates that the contribution of this paper may better be discussed in a format that has less restrictive page constraints. There are reasons why papers have page limits and I have the impression that for this work the authors are trying to circumvent them by providing a lot of material in the Appendix.

(3) The paper does not discuss any limitations of the introduced method.

Therefore, I don't think the paper is ready for publication.

---

> ### Author Response · Authors · 2021-11-23
> **Author Response to Reviewer dYcg [2/2]**
>
> ## Breif List of Revisions
> ### Common Modification
> 1. Replace "system" by "domain" to denote a specific configuration of data, e.g. FluidFall and FluidShake are two different domains.
> 2. Replace "GCN" by "GNN".
>
> ### Abstract
> 1. Limit the methods discussed in our paper to learning-based methods.
> 2. Add explanations for 'material semantics', 'particle tokens', and 'interaction tokens'.
>
> ### Introduction
> 1. Add explanations for notions, such as 'traditional simulators', 'particle interactions', 'material semantics', 'interaction tokens'.
> 2. Clearer descriptions of our methods.
>
> ### Related Work
> 1. Correct the order of 'CConv' and 'GNS'.
> 2. Add reference of 'COPINGNet'.
>
> ### Methodology
> 1. Add explanations for notions, such as 'domain-specific semantics', 'domain-agnostic semantics'.
> 2. Clearer descriptions of our methods.
> 3. Correct the formulation of WMSE in Equation 16.
>
> ### Experiments
> 1. Update GNS related performances and analysis.
> 2. Add 'sensitiveness of SiT to radius $R$' and 'effectiveness of abstract tokens' into ablation studies.
> 3. Add results for DPI-Net on few shorts learning.
>
> ### Appendix
> 1. Update GNS related performances and analysis.
> 2. Add details about implementations in Appendix 6.1.3.

---

> ### Author Response · Authors · 2021-11-23
> **Author Response to Reviewer dYcg [1/2]**
>
> We thank reviewer for the valuable feedback. We have revised the paper to reflect the feedback, where revision is marked with magenta color.
> Below we address each concern in detail.
>
> ## Q1: Unclear notions and descriptions.
> Thanks for your advice. We have modified our revision accordingly. Detailed modifications:
> 1. As mentioned in the abstract, we focus on learning-based simulators in this paper.
> 2. We add further explanations on 'material semantics', 'particle tokens', and 'interaction tokens' in abstract.
> 3. We add examples and references for "traditional simulators" in Section 1 Paragraph 1.
> 4. We replace 'same pipeline' in 'Consequently, they follow the same pipeline ...' by clearer descriptions in Section 1 Paragraph 1.
> 5. We tone-down the statement of "where they can robustly estimate the dynamics of a system with varying number and configuration of particles" in Setion 1 Paragraph 1.
> 6. We add an example for "rich semantics of particle interactions" in Section 1 Paragraph 1.
> 7. We replace "deploy an abstract token" by clearer descriptions in Section 1 Paragraph 3.
> 8. We give a high-level understandings of our contributions in Section 1 Paragraph 3 and details in Section 1 Paragraph 4. We replace "good balance between efficiency and complexity" by clearer descriptions. We explain the concepts at their first appearances. Besides, we replace "system" by "domain" to denote a specific settings of environment. The references of the notions are as follows:
>
>    8.1. Material semantics(characteristics): such as viscosity or plastic deformations. (Abstract; Section 1 Paragraph 2)
>
>    8.2. Particle interaction: the influence of action-reaction forces, such as the collisions. (Section 1 Paragraph 1)
>
>    8.3. Interaction semantics: such as how the particle is restored after deformations. (Section 1 Paragraph 3)
>
>    8.4. Interaction tokens: high-dimensional representations for interactions. (Section 1 Paragraph 3)
>
>    8.5. Abstract tokens: a high-dimensional embedding for each type of material to capture material semantics. (Section 1 Paragraph 3)
>
>    8.6. Domain-specific semantics: such as the shape of rigid cubic.  (Section 1 Paragraph 2; Section 3.3 Paragraph 1)
>
>    8.7. Domain-agnostic semantics: such as the stickiness attribute of materials. (Section 3.3 Paragraph 1)
> 9. As we mentioned "learning-based methods" in abstract, 'The goal of a simulator is to learn a model ...' refers to learning-based simulators in Section 3.1.
> 10. We replace the sentence 'a selective mechanism is thus in great need to help the simulator' according to your advice in Section 3.2 Paragraph 1.
> 11. More details about why transformer is a good backbone are provided in Section 3.2 Paragraph 2.
> 12. We provide more details of the redundant computations for all interactions in Section 3.2 Paragraph 3.
> 13. "Rice semantics of particle interactions" (particle interaction) is explained at Section 1 Paragraph 1.
> 14. "The semantics of a system" refers to "domain-specific semantics". And we add explanations right after this sentence in Section 3.3 Paragraph 1.
> 15. We provide examples to explain "domain-specific semantics" and "domain-agnostic semantics" in Section 3.3 Paragraph 1.
> 16. "Domain-agnostic semantics" is explained in Section 3.3 Paragraph 1.
> 17. As suggested, we add reference of 'Accurately Solving Rod Dynamics with Graph Learning' in Related Work.
>
>
> ## Q2: Appendix provide additional material
>
> A:
> At first, the appendix provides **no new content** beyond those covered by the main paper. It is sufficient to read only the main paper as it is self-contained. Appendix provides only optional and supplementary content as allowed by the author guidelines.
>
> Second, in terms of the Table 4 mentioned in your comments, we already provide the same type of study and relevant discussion in Table 2 and Section 4.2 of the main paper. Table 2 compares SiT to DPI-Net in various generalized unseen settings, as DPI-Net is the representative method applicable to all environments.
>
> ## Q3: Limitations
>
> A:
> At first, although SiT provides learnable abstract tokens that captures material-aware semantics, they may be hard to interpret in order to adjust a specific aspect of the semantics.
>
> Secondly, currently SiT considers only pair-wise interactions. The joint interactions of more than two particles can also be considered.

---

### Official Review · Reviewer_DCt1 · 2021-11-03

**Correctness:** 3
**Technical Novelty And Significance:** 3
**Empirical Novelty And Significance:** 3
**Recommendation:** 6
**Confidence:** 3

**Main Review:**

SiT uses the Transformer architecture to model particle dynamics. In contrast to a vanilla Transformer, SiT incorporates interaction tokens and abstract tokens to better model particle interactions and material properties, respectively. SiT is evaluated in four simulated environments: FluidFall, FluidShake, BoxBath, RiceGrip, though it would be nice to see environments with more non-liquid multi-object interactions.

**Summary Of The Paper:**

The paper proses the Simulation Transformer (SiT) to simulate particle dynamics, using the Tranformers' attention mechanism to attend to critical particle interactions. They demonstrate qualitative and quantitative improvements over prior work with various architectures in multiple environments.

**Summary Of The Review:**

SiT provides a novel Transformer-based architecture for predicting particle dynamics that outperforms previous graph-convolutional and Transformer-based architectures.

---

> ### Author Response · Authors · 2021-11-23
> **Author Response to Reviewer DCt1 [1/1]**
>
> We thank reviewer for the valuable feedback. We have revised the paper to reflect the feedback, where revision is marked with magenta color.
> Below we briefly list the revisions in our paper.
>
> ## Breif List of Revisions
> ### Common Modification
> 1. Replace "system" by "domain" to denote a specific configuration of data, e.g. FluidFall and FluidShake are two different domains.
> 2. Replace "GCN" by "GNN".
>
> ### Abstract
> 1. Limit the methods discussed in our paper to learning-based methods.
> 2. Add explanations for 'material semantics', 'particle tokens', and 'interaction tokens'.
>
> ### Introduction
> 1. Add explanations for notions, such as 'traditional simulators', 'particle interactions', 'material semantics', 'interaction tokens'.
> 2. Clearer descriptions of our methods.
>
> ### Related Work
> 1. Correct the order of 'CConv' and 'GNS'.
> 2. Add reference of 'COPINGNet'.
>
> ### Methodology
> 1. Add explanations for notions, such as 'domain-specific semantics', 'domain-agnostic semantics'.
> 2. Clearer descriptions of our methods.
> 3. Correct the formulation of WMSE in Equation 16.
>
> ### Experiments
> 1. Update GNS related performances and analysis.
> 2. Add 'sensitiveness of SiT to radius $R$' and 'effectiveness of abstract tokens' into ablation studies.
> 3. Add results for DPI-Net on few shorts learning.
>
> ### Appendix
> 1. Update GNS related performances and analysis.
> 2. Add details about implementations in Appendix 6.1.3.

---

### Official Review · Reviewer_yAKg · 2021-11-04

**Correctness:** 3
**Technical Novelty And Significance:** 2
**Empirical Novelty And Significance:** 2
**Recommendation:** 6
**Confidence:** 3

**Main Review:**

**Strengths:**
1. Paper is well written, well organized, and narrative is coherent.

2. Experimentation for comparisons with previous state of the art models are convincing as authors compare with 4 recent related approaches and demonstrate that SiT improves performance over other models for fluid simulation in different contexts using standard datasets.

3. Specifically, an interesting contribution of the paper is the use of explicit (disentangled) "abstract tokens" that encode material specific properties. The authors demonstrate (through multi-material simulations like a solid cube floating in a fluid) that the SiT method is able to leverage the material specific properties learned in the "abstract tokens" to better simulate the properties of the solid interacting with the fluid. For example: other methods (without abstract tokens) which don't explicitly differentiate between the solid and the fluid simulate the solid disintegrating and merging with the flowing fluid but SiT maintains the solid shape and is able to simulate the motion of the solid over the moving fluid.

4. Demonstrated SiT models are much more light-weight (i.e., far fewer parameters). Training methodology further explicitly prioritizes learning properties for different material types by adopting a weighted MSE loss.


**Weaknesses:**

1. Some of the initial claims made by the authors of disadvantages of GCNs (as they presuppose particle interaction neighborhoods where interaction occurrs with all particles in neighborhood) seem to be part of the proposed method as well? For example, even in the SiT model, the particle token interactions are defined by a window function governed by a radius `R`, isn't the particle token interacting with all other particles within radius `R`?

2. Incomplete ablation analysis, authors do not demonstrate the importance of abstract tokens.

3. In the "few shot generalization" case, it is unclear what the goal of the experiment is? Also, in in Fig. 5b what does "robust" mean? i.e., What does it mean to claim that the SiT model (i.e., without abstract tokens) is "robust even at 60% training data" ? Secondly, the claim of "few-shot" learning while the model uses 60% of training data is questionable.


**Questions:**

1. How is the parameter governing the extent of the particle window to inform particle connections (`R`) selected?

2. How sensitive is SiT to the selection of `R` ?

3. Also, in the case of pair-wise interaction components i.e., particle interaction tokens, how is setting allowable connections (using Eq 5) any different from setting connections in the GCN works (that authors describe as inferior) earlier in their paper?

4. Authors claim that GCN based mechanisms are unable to provide selection mechanisms? Would employing attention in the GCN context not serve a a selection mechanism?

5. All abstract tokens of a particular material interact with each other? How scalable is this operation for large fluid simulations?

**Summary Of The Paper:**

In this paper, authors propose Simulation Transformer (SiT), a transformer based approach for particle-based fluid simulations (in contrast to all the existing approaches which are overwhelmingly based on Graph Convolutional Networks). Specifically, in their paper the authors augment the vanilla transformers by incorporating sub-networks for richer modeling particle interactions (i.e., as opposed to particle interactions being modeled as dot products and being reduced to a single number, sub-networks allow richer modeling of interactions), as well as material specific properties. Authors evaluate the proposed SiT model on diverse environments and compare against several state of the art fluid simulation models and also demonstrate generalization across different materials.

**Summary Of The Review:**

The paper makes a contribution of being the first work to employ transformer to model particle based simulations where the model explicitly accounts for particle, particle interaction as well as material properties during the simulation. The authors have performed a rigorous comparison with state of the art models and the achieved improvements and the explanations rendered for the improvements are sensible. Also, authors demonstrate also through qualitative and quantitative comparisons, the generalizability of the SiT model in various multi-material settings which are also convincing.

---

> ### Author Response · Authors · 2021-11-23
> **Author Response to Reviewer yAKg [2/2]**
>
> ## Q5: How sensitive is SiT to $R$
>
> A:
>
> | Methods \ MSE(1e-2) | $R=0.07$           | $R=0.08$           | $R=0.09$           |
> | ------------------- | ------------------ | ------------------ | ------------------ |
> | DPI-Net             | $2.60\pm0.56$      | $1.43\pm0.52$      | $1.66\pm0.48$      |
> | SiT                 | $1.38\pm0.36$      | $1.08\pm0.36$      | $1.37\pm0.35$      |
>
> The results of different $R$ values on FluidShake are shown above.
>
> When $R$ increases, the performance is negatively affected as more redundant or even false interactions are considered. Compared to DPI-Net, SiT can still achieve better results and maintain smaller standard deviation, which indicates that SiT is not very sensitive to the change of $R$ within a small range.
>
> When $R$ decreases, the performance becomes worse due to the lack of important interactions. Nevertheless, SiT also achieves better results than DPI-Net.
>
> In conclusion, SiT is more robust within a small range of $R$ compared to DPI-Net. And SiT with different values of $R$ all outperform DPI-Net with the best setting ($R=0.08$).
>
> It is worth noting that our observation presented here is different from what was observed by the authors of DPI-Net and GNS. In their paper, MSEs keep decreasing as the value of $R$ increases. This is because we conduct the experiments on domains that are different from them.
>
> ## Q6: The same way to set connections as GCNs
>
> A:
> Admittedly, we use the same window function (Eq.5) as GCN-based methods for our current settings. As discussed in Q1, the limitation of GCN-based methods is not the window function but the lack of a selective mechanism. In contrast, SiT provides an effective selective mechanism based on the self-attention of Transformer.
>
> Moreover, SiT is more flexible for extensions, where it can adjust Eq.5 and Eq.8 to respectively include more complex interactions such as the ones involve more than two particles and use advanced formuations for updating interaction tokens.
>
>
> ## Q7: GCN can also provide a selection mechanism.
>
> A: At first, current GCN-based methods didn't adopt any attention modules, leading to no selective mechanism over interactions. We emphasize that they did not take the selective mechanism into consideration.
>
> As suggested, we also conduct the experiment where we equip GNS with an attention module thus providing it with a selective mechanism.
> Below we show the results of GNS, GNS+attention, and our SiT on the FluidShake environment. Hyper-parameters are chosen to ensure a fair comparison.
>
> | Methods \ FluidShake  | MSE(1e-2)          | Parameters         |
> | --------------------- | ------------------ | ------------------ |
> | GNS           | $1.45\pm0.55$      | 1.59M              |
> | GNS + attention       | $1.43\pm0.39$      | 1.59M              |
> | SiT                   | $1.08\pm0.36$      | 0.77M              |
>
> As shown by the results, although with a selective mechanism, GNS achieves lower MSEs and smaller standard deviations, SiT still achieves superior results with less parameters, indicating the effectiveness of our selective mechanism. Intuitively, the self-attention module as well as the interaction tokens provide more room for SiT to select out important interactions.
>
> ## Q8: All abstract tokens of a particular material interact with each other? How scalable is this operation for large fluid simulations?
>
> A: At first, each type of material has **only one single abstract token**, which interacts with all particles belonging to that material.
> There are no interactions between abstract tokens of different materials. Therefore, the abstract tokens only introduce small computational overhead, so that SiT with abstract tokens share the similar scalability with SiT without abstract tokens.
>
>
> ## Breif List of Revisions
> ### Common Modification
> 1. Replace "system" by "domain" to denote a specific configuration of data, e.g. FluidFall and FluidShake are two different domains.
> 2. Replace "GCN" by "GNN".
>
> ### Abstract
> 1. Limit the methods discussed in our paper to learning-based methods.
> 2. Add explanations for 'material semantics', 'particle tokens', and 'interaction tokens'.
>
> ### Introduction
> 1. Add explanations for notions, such as 'traditional simulators', 'particle interactions', 'material semantics', 'interaction tokens'.
> 2. Clearer descriptions of our methods.
>
> ### Related Work
> 1. Correct the order of 'CConv' and 'GNS'.
> 2. Add reference of 'COPINGNet'.
>
> ### Methodology
> 1. Add explanations for notions, such as 'domain-specific semantics', 'domain-agnostic semantics'.
> 2. Clearer descriptions of our methods.
> 3. Correct the formulation of WMSE in Equation 16.
>
> ### Experiments
> 1. Update GNS related performances and analysis.
> 2. Add 'sensitiveness of SiT to radius $R$' and 'effectiveness of abstract tokens' into ablation studies.
> 3. Add results for DPI-Net on few shorts learning.
>
> ### Appendix
> 1. Update GNS related performances and analysis.
> 2. Add details about implementations in Appendix 6.1.3.

---

> ### Author Response · Authors · 2021-11-23
> **Author Response to Reviewer yAKg [1/2]**
>
> We thank reviewer for the valuable feedback. We have revised the paper to reflect the feedback, where revision is marked with magenta color.
> Below we address each concern in detail.
>
> ## Q1: SiT also interacts with all neighbors using a radius $R$, which is similar to GCN-based methods.
>
> A:
> It is worth noting that we didn't regard the usage of $R$ as a limitation of GCN-based methods.
> In the second paragraph of Section 1, we emphasize that the limitation of existing GCN-based methods is that they does not provide a selective mechanism, where they treat all interactions equally by directly summing the edge features for further processing.
>
> In contrast, SiT resort to the self-attention modules to select important interactions. Moreover, SiT also utilizes abstract tokens to capture essential material-aware semantics. The selective mechanism enabled by the self-attention modules and the abstract tokens not only improve the performance of SiT, but also make SiT more robust on different domains.
>
> While current GCN-based methods have no selective mechanism, as pointed out by the reviewer, there are GCN variants that provide an attention module that can serve as the selective mechanism. To this end, as suggested we equip our baseline, GNS, with an attention module, and evaluate it on the environment FluidShake. Hyper-parameters are chosen to ensure a fair comparison. Please refer to the response of Q7 for more discussion. The results are shown in the following:
>
> | Methods \ FluidShake  | MSE(1e-2)          | Parameters         |
> | --------------------- | ------------------ | ------------------ |
> | GNS                   | $1.45\pm0.55$      | 1.59M              |
> | GNS + attention       | $1.43\pm0.39$      | 1.59M              |
> | SiT                   | $1.08\pm0.36$      | 0.77M              |
>
>
> ## Q2: Incomplete ablation analysis, authors do not demonstrate the importance of abstract tokens.
>
> A: In table 1 we compare the performance of our method with and without abstract tokens (respectively dubbed as SiT and SiT+). And the relevant discussion is included in Section 4.1.
>
> In addintion, we add extra ablation study about abstract tokens in Section 4.3 and report the results in Table 4, suggesting that abstract tokens are able to learn the semantics of materials. Please refer to Section 4.3 for more details.
>
> ## Q3: The goal of "few shot generalization" is not clear.
>
> A:
> First, we aim to verify how sensitive SiT is to the amount of training data. "Robust" means whether the model can maintain its accuracy and predict perceptually faithful results with less training data. When trained on 60% of data, SiT can achieve similar MSE results ($1.62 \pm 0.44$) comparing with DPI-Net ($1.43\pm0.52$), which is trained on the whole dataset. We further compare with DPI-Net when trained with same amount of data:
>
> | Methods \ MSE(1e-2) | Train on 20%       | Train on 40%       | Train on 60%       | Train on 80%       | Train on 100%       |
> | ------------------- | ------------------ | ------------------ | ------------------ | ------------------ | ------------------- |
> | DPI-Net             | $3.00\pm1.21$      | $2.43\pm0.57$      | $1.88\pm0.44$      | $2.04\pm0.67$      | $1.43\pm0.52$       |
> | SiT                 | $4.52\pm1.24$      | $3.03\pm0.44$      | $1.62\pm0.44$      | $1.39\pm0.70$      | $1.08\pm0.36$       |
>
> The results suggest that with no less than 60% of training data, SiT is more robust than DPI-Net and still achieves good results. When there are fewer training examples, both SiT and DPI-Net fail to predict faithful rollouts, leading to larger MSEs.
>
> ## Q4: How R is selected
>
> A: We use the same setting as in previous work. In DPI-Net and GNS, they both adopt $R=0.08$ on the data generated by PyFleX.
> To ensure a fair comparison, we also use $R=0.08$ by default. The effect of different $R$ values is included in the response of Q5.

---

### Decision · Program_Chairs · 2022-01-20

**Decision:**

Reject

**Comment:**

The submission received split reviews: two reviewers recommended weak accepts, and the other two weak rejects.  The AC went through the reviews, responses, and discussions carefully.  The AC agrees that this paper is well-written and has demonstrated the possibility of using transformers for particle-based physical simulation.  The AC also believes that the authors have addressed the concerns of reviewer dYcg, despite that the reviewer didn't engage in the discussions.  The contributions are however not most exciting, and none of the reviewers would like to champion the submission.

Further, the AC agrees with the knowledgeable and responsible reviewer ZsKn that the presentation and experiments can be better positioned to highlight the key contribution. As reviewer ZsKn has summarized, it's recommended that "the authors took the approach of integrating the different parts of the newly proposed layer into existing architectures (possibly including non-simulation settings), and try to understand better that way how the new layer may help in a more apples-to-apples comparison."

The recommendation is reject, and the authors are encouraged to revise the paper for the next venue.